# `Multi`$^2$: Hierarchical Multi-Agent Decision-Making with LLM-Based Agents in Interactive Environments

Sangeun Park [1]   Minhae Kwon [1]

## Abstract

A central goal of large language model (LLM) research is to build agentic systems that can plan, act, and adapt through sustained interaction with dynamic environments. While recent LLM-based agents exhibit impressive contextual reasoning, their long-horizon decision-making remains fragile, often suffering from *objective drift*, where goals and plans drift over extended interactions. We introduce `Multi`$^2$, a hierarchical multi-agent decision-making framework that explicitly decomposes agent behavior into complementary roles. A high-level agent (`System 1`) focuses on context-aware sub-goal generation using supervised fine-tuning (SFT), while a low-level agent (`System 2`) executes atomic actions through offline-to-online reinforcement learning (RL) in interactive environments. This separation enables stable long-horizon control, mitigates objective drift, and allows efficient adaptation. Across diverse interactive environments, `Multi`$^2$ consistently outperforms strong agentic baselines, demonstrating improved robustness and coordination in multi-turn interaction. Beyond performance, we introduce and release three hierarchical benchmark datasets, filling a long-standing gap in training and evaluating hierarchical decision-making for LLM-based agents.

Project Page: https://park-sangeun. github.io/Multi-Square/

## 1. Introduction

A central goal in agentic artificial intelligence (AI) is to build LLM-based agents that can solve complex tasks through closed-loop interaction with dynamic environments, where

[1]Department of Electrical and Computer Engineering, Sungkyunkwan University, Republic of Korea. Correspondence to: Minhae Kwon <minhae.kwon@skku.edu>.

*Proceedings of the 43$^{rd}$ International Conference on Machine Learning*, Seoul, South Korea. PMLR 306, 2026. Copyright 2026 by the author(s).

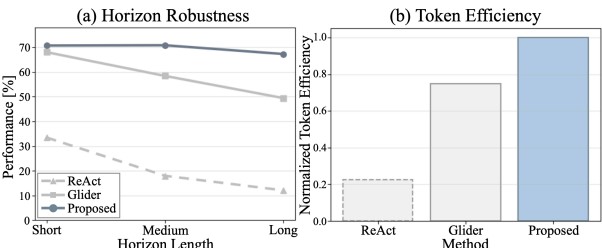

*Figure 1.* Challenges in long-horizon interaction on the Science-World using Llama-3.1 8B backbone. We compare `ReAct` [14] (prompt-based, non-hierarchical), `Glider` [15] (hierarchical fine-tuning baseline), and our method (`Multi`$^2$). (a) **Horizon Robustness:** performance as a function of horizon length, where longer horizons induce larger degradation for baselines. (b) **Token Efficiency:** normalized inference-time token efficiency (higher is better), computed as performance/tokens and normalized such that the proposed method equals 1.0.

each action changes the world state and must be followed by decisions conditioned on new observations [1, 2, 3, 4]. Such settings inherently require multi-turn, long-horizon decision-making [5, 6, 7, 8]. However, long-horizon interaction remains brittle [9, 10]. Agents often exhibit *objective drift*—gradually deviating from the intended goal as interaction unfolds—and become token-inefficient by relying on ever-growing interaction histories to maintain intent [11, 12]. In practice, objective drift is not merely a planning failure, but an interaction-level phenomenon where small execution errors accumulate over time, progressively misaligning the agent's state with the original intent [13, 14]. This raises the question:

> *How can we build interactive LLM-based agents that reliably and efficiently solve long-horizon tasks?*

Recent studies mitigate such drift by decomposing long tasks into sub-goals via hierarchical planning [15, 16]. Yet, long-horizon interaction is not fully solved: decomposition can reduce planning failures, but execution can still break down over extended rollouts due to compounding errors and constraint-heavy actions [17, 18]. Moreover, many hierarchical agents implement role specialization primarily through prompting, which can blur as context accumulates, leaving agents vulnerable to constraint violations and loops [19]. These observations reveal a more fundamental limitation: while hierarchical decomposition improves plan-

ning, existing approaches fail to ensure stable execution, as the executor is not explicitly trained to correct compounding errors during interaction [20, 21]. Accordingly, we argue that reliable multi-turn interaction requires (1) explicit task-level decomposition to preserve intent, (2) robust action execution that improves through interaction with the environment, and (3) token-efficient interaction.

To quantify these challenges, Figure 1 compares ReAct [14], Glider [15], and Multi² in ScienceWorld [22] in terms of (a) horizon robustness and (b) token efficiency. Glider improves over ReAct via hierarchical decomposition, but its performance still degrades as horizons lengthen, indicating that long-horizon brittleness and higher token costs remain under multi-turn interaction. In contrast, Multi² sustains higher performance at longer horizons while achieving better token efficiency, underscoring the need for reliable and efficient interactive LLM-based agents.

We therefore propose **Multi²**, a hierarchical multi-agent framework that decouples task decomposition (System 1) and action execution (System 2) through role specialization. Specifically, System 1 generates context-aware sub-goals that preserve global intent, while System 2 executes atomic actions in the dynamic environment, enabling selective invocation for token-efficient interaction. We train the two systems with a role-aligned pipeline: System 1 via SFT for structured sub-goal planning, and System 2 via offline-to-online RL to initialize stably from offline data and continually self-improve through online interaction.

In summary, our contributions are as follows: (1) We propose **Multi²**, a role-specialized hierarchical training-and-adaptation framework that decouples sub-goal planning and action execution as different optimization problems for interactive environments. This design improves goal consistency and mitigates objective drift in long-horizon interaction. (2) We introduce a principled offline-to-online RL objective for System 2, combining policy-anchored offline learning with Kullback-Leibler (KL)-regularized online refinement for stable self-improvement with a novel loss design. (3) We release well-structured hierarchical datasets for training and evaluating interactive LLM-based agents across multiple environments (see Section 5.1). (4) Multi² consistently outperforms strong baselines in performance, token efficiency, and long-horizon robustness.

## 2. Related Works

**Hierarchical Structure for Interactive LLM-Based Agents.** There has been growing interest in applying AI systems to interactive environments [23, 24, 25, 26], where agents must perceive changing states, make sequential decisions, and adapt their behavior over time [27, 28, 29]. In this context, recent works have explored LLM-based agents that interact with external environments (e.g., embodied simulators and text games) [1, 30, 31, 32]. While such settings demand long-horizon, multi-turn interaction, many existing systems primarily target short-horizon tasks, leaving robust decision-making as an open challenge [11, 12, 33]. Common approaches build on sequential decision-making paradigms, such as ReAct [14] and Reflexion [34], which often rely on long interaction histories and can be token-inefficient. Moreover, these approaches are susceptible to objective drift when the initial intent or plan falls outside the context window [35, 36, 37, 38].

To address these challenges, recent studies introduce hierarchical structures that decouple high-level task decomposition from low-level execution [16, 15, 39, 40, 41, 42]. For instance, ADaPT [16] proposes a hierarchical approach that explicitly plans and decomposes complex sub-tasks. Glider [15] introduces a hierarchical RL framework that leverages intermediate sub-goals to ease credit assignment in long-horizon settings. However, these approaches primarily focus on decomposition, leaving execution robustness largely underexplored. In particular, the executor is typically not optimized to correct compounding errors during interaction, which limits their effectiveness in long-horizon settings [17, 18]. Our work addresses this gap by explicitly coupling hierarchical decomposition with a learning-based execution policy that improves through interaction.

**Offline-to-Online Reinforcement Learning.** A standard approach for adapting LLM-based agents is SFT [43], but it can overfit to domain-specific trajectories and generalize poorly beyond the training data [44, 45, 46, 47]. RL provides an alternative by improving behavior through interaction rather than blind imitation [48, 49, 50, 51, 52, 53]. However, purely online RL tends to be sample-inefficient and brittle, and can become costly due to unsafe exploration during rollouts [54, 55, 56, 57, 58, 59], whereas purely offline RL is constrained by dataset quality and distributional coverage [60, 61, 62, 63, 64], limiting improvements beyond the behavior policy [65, 66, 67, 68].

These trade-offs are exacerbated for long-horizon LLM-based agents, where dynamic states and constraint-heavy actions are difficult to cover with static datasets. Recent work has explored offline-to-online RL, using offline data to stably initialize a policy without online interaction, then improving it through online interaction [69, 70, 71, 72, 73, 74]. While prior offline-to-online RL methods focus on stabilizing policy improvement, they are not tailored to the structured execution requirements induced by hierarchical agents. In contrast, we design an objective that explicitly aligns with the role of the executor, enabling stable and sample-efficient learning, while supporting continual self-improvement.

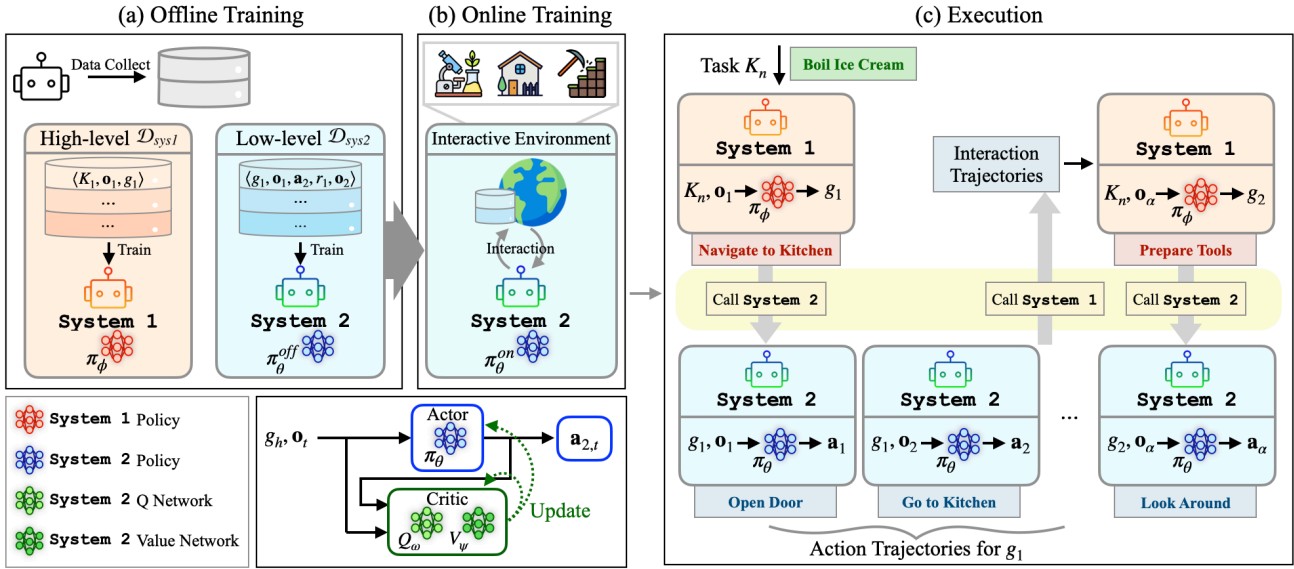

*Figure 2.* Overview of Multi². (a) **Offline Training:** System 1 (high-level agent) is trained via SFT on the high-level dataset $\mathcal{D}_{sys1}$ to generate sub-goals, while System 2 (low-level agent) is initialized via offline RL on the low-level dataset $\mathcal{D}_{sys2}$ using an actor-critic objective. (b) **Online Training:** starting from the offline policy, System 2 continues self-improvement through interaction, updating its policy while retaining and augmenting the replay buffer. (c) **Execution:** at inference time, System 1 proposes context-aware sub-goals and System 2 executes atomic actions conditioned on each sub-goal; System 1 is re-invoked when a sub-goal is achieved.

## 3. Problem Setups

### 3.1. Problem Setup for LLM-Based Agents

We denote the high-level agent as System 1 for context-aware planning and the low-level agent as System 2 for action execution. Each agent is modeled as a partially observable Markov decision process tailored to its role. At each high-level step, System 1 takes a task description $K_{n \in N}$ and the observation $\mathbf{o}_t$ as input and outputs a sub-goal $g_h$. Conditioned on $\mathbf{o}_t$ and $g_h$, System 2 executes an atomic action $\mathbf{a}_t$ to accomplish the sub-goal $g_h$.

### 3.2. Sequential Decision-Making Problem

We formulate the hierarchical LLM-based agents as a partially observable Markov decision process. It is defined by the tuple $\langle \mathcal{S}, \mathcal{A}, \mathcal{O}, \mathcal{T}, \Omega, \mathcal{R}, \gamma \rangle$, where $\mathbf{s}_t \in \mathcal{S}$ is a state, $\mathbf{a}_t \in \mathcal{A}$ is an action, $\mathbf{o}_t \in \mathcal{O}$ is an observation. The state transition probability is given by $\mathcal{T} : \mathcal{S} \times \mathcal{A} \rightarrow \mathcal{S}$, the observation probability by $\Omega : \mathcal{S} \rightarrow \mathcal{O}$, the reward function by $\mathcal{R} : \mathcal{S} \times \mathcal{A} \times \mathcal{S} \rightarrow \mathbb{R}$, and $\gamma \in [0,1)$ denotes a temporal discount factor.

The objective of RL is to learn a policy that maximizes the expected cumulative return $\mathcal{J}_\pi = \mathbb{E}_\pi \left[ \sum_t \gamma^t r_t \right]$, where $r_t = \mathcal{R}(\mathbf{s}_t, \mathbf{a}_t, \mathbf{s}_{t+1})$. We adopt an actor-critic structure, where the actor selects an action $\mathbf{a}_t$ according to the policy $\pi_\theta(\mathbf{a}_t \mid \mathbf{o}_t)$, while the critic estimates the action-value $Q_\omega(\mathbf{o}_t, \mathbf{a}_t) = \mathbb{E}_\pi \left[ \sum_f \gamma^f r_{t+f} \mid \mathbf{o}_t, \mathbf{a}_t \right]$ and the state-value

$V_\psi(\mathbf{o}_t) = \mathbb{E}_{\pi_\theta}[Q_\omega(\mathbf{o}_t, \mathbf{a}_t)]$. Accordingly, the advantage function is defined as $A(\mathbf{o}_t, \mathbf{a}_t) = Q_\omega(\mathbf{o}_t, \mathbf{a}_t) - V_\psi(\mathbf{o}_t)$, which measures how much better a specific action is than the expected policy value at $\mathbf{o}_t$. Within the actor-critic framework, the actor updates its policy using the value estimates from the critic, while the critic learns by minimizing the temporal-difference (TD) error.

## 4. The Proposed Multi² Framework

We propose Multi², a hierarchical multi-agent decision-making framework for long-horizon interactive tasks (Figure 2). Multi² couples hierarchical control with an offline-to-online RL training pipeline. Given a task $K_n$, System 1 generates a sub-goal $g_h \sim \pi_\phi(\cdot \mid \mathbf{o}_t; K_n)$, which specifies a sub-task for System 2. Conditioned on the current observation and sub-goal, System 2 selects an atomic action $\mathbf{a}_t \sim \pi_\theta(\cdot \mid \mathbf{o}_t; g_h)$ to accomplish $g_h$.[1] Once $g_h$ is achieved (or terminated), System 2 queries System 1 to generate the next sub-goal $g_{h+1}$, and the process repeats.

### 4.1. Hierarchical Dataset Construction

For training the agents, we prepare role-specific hierarchical datasets. Based on the input–output requirements of each agent, the overall dataset is partitioned into $\mathcal{D}_{sys1}$ and $\mathcal{D}_{sys2}$.

---

[1]Here, $\phi$ and $\theta$ denote the parameters of System 1 and System 2, respectively.

The dataset for System 1 is defined as follows.

$$\mathcal{D}_{sys1} = \left\{ \left( K_n, \{(\mathbf{o}_h, g_h)\}_{h=1}^{H} \right) \right\}_{n=1}^{|\mathcal{D}_{sys1}|} \quad (1)$$

In (1), $K_n$ denotes the natural-language task description, and $\{(\mathbf{o}_h, g_h)\}_{h=1}^{H}$ denotes a sequence of observation–sub-goal pairs, where $H$ is the number of sub-goals.

The dataset for System 2 is defined as follows.

$$\mathcal{D}_{sys2} = \left\{ \left( g^{(i)}, \ (\xi_t^{(i)})_{t=1}^{M^{(i)}} \right) \right\}_{i=1}^{|\mathcal{D}_{sys2}|}$$
$$\xi_t^{(i)} = \left( \mathbf{o}_t^{(i)}, \ \mathbf{a}_t^{(i)}, \ r_t^{(i)}, \ \mathbf{o}_{t+1}^{(i)} \right) \quad (2)$$

In (2), $g^{(i)}$ denotes the $i$-th sub-goal provided by System 1, and $(\xi_t^{(i)})_{t=1}^{M^{(i)}}$ denotes the ordered sequence of low-level transitions collected while the sub-goal is active. Each transition is defined as $\xi_t^{(i)} = (\mathbf{o}_t^{(i)}, \mathbf{a}_t^{(i)}, r_t^{(i)}, \mathbf{o}_{t+1}^{(i)})$, where $M^{(i)}$ represents the number of low-level transitions associated with the $i$-th sub-goal instance. By separating datasets for sub-goal generation and action control, this formulation supports role-specialized training for hierarchical decision-making. For clarity, the prompt-formatted dataset construction is provided in Appendix A.

## 4.2. Role-Specialized Offline Training for System 1 and System 2

To train two systems efficiently, we adopt role-specific objectives aligned with their roles. System 1 focuses on semantic, context-aware decision-making, while System 2 is responsible for atomic action selection. The overall procedure, including the training of System 1 and System 2, is summarized in Algorithm 1.

**System 1 Training with SFT.** System 1 serves as a high-level planner that generates sub-goals for each task $K_n$. Since sub-goal generation primarily requires consistent semantic planning, we train System 1 with SFT to provide a stable supervision signal and encourage task-level consistency. Specifically, we fine-tune the policy $\pi_\phi$ by behavior cloning on the high-level dataset $\mathcal{D}_{sys1}$.

$$\mathcal{L}_\phi = -\mathbb{E}_{(K_n, \mathbf{o}_t, g_h) \sim \mathcal{D}_{sys1}} [\log \pi_\phi(g_h \mid \mathbf{o}_t; K_n)] \quad (3)$$

In (3), $\mathcal{L}_\phi$ is the standard cross-entropy objective that encourages System 1 to imitate expert sub-goals in $\mathcal{D}_{sys1}$, promoting task-level consistency in long-horizon planning.

**System 2 Training with Offline RL.** System 2 executes atomic actions $\mathbf{a}_t$ to accomplish the current sub-goal $g_h$ in interactive environments and capture subtle state changes over long horizons. To this end, we introduce an offline RL objective for System 2 that stabilizes training by learning an

actor and critic using the training dataset. We optimize the policy $\pi_\theta$ using the Q-function $Q_\omega$ and the value function $V_\psi$. First, the Q-function $Q_\omega$ is trained by minimizing the TD error, as follows.

$$\mathcal{L}_\omega = \mathbb{E}_{(g_h, \mathbf{o}_t, \mathbf{a}_t, r_t, \mathbf{o}_{t+1}) \sim \mathcal{D}_{sys2}} \Big( r_t + \gamma V_\psi(\mathbf{o}_{t+1}; g_h)$$
$$- Q_\omega(\mathbf{o}_t, \mathbf{a}_t; g_h) \Big)^2 \quad (4)$$

In (4), the term $r_t + \gamma V_\psi(\mathbf{o}_{t+1}; g_h)$ serves as the TD target, encouraging $Q_\omega$ to estimate the expected return of taking $\mathbf{a}_t$ under sub-goal $g_h$ using only dataset-supported transitions. Here, $Q_\omega$ estimates the expected return for taking action $\mathbf{a}_t$ under sub-goal $g_h$, while $V_\psi$ estimates the expected return from the observation conditioned on $g_h$. This in-distribution (ID) value learning helps stabilize offline training by reducing reliance on out-of-distribution (OOD) actions that are not supported by the dataset.

The value function $V_\psi$ is updated using expectile regression.

$$\mathcal{L}_\psi = \mathbb{E}_{(g_h, \mathbf{o}_t, \mathbf{a}_t, r_t, \mathbf{o}_{t+1}) \sim \mathcal{D}_{sys2}} \Big[ L^\tau(A(\mathbf{o}_t, \mathbf{a}_t)) \Big] \quad (5)$$

In (5), $L^\tau(u) = |\tau - \mathbb{1}\{u < 0\}| u^2$ is the expectile loss and $\mathbb{1}\{\cdot\}$ is the indicator function.

The actor is updated using both the Q-function $Q_\omega$ and the value function $V_\psi$, which is as follows.

$$\mathcal{L}_\theta^{\text{offline}} = -\mathbb{E}_{(g_h, \mathbf{o}_t, \mathbf{a}_t) \sim \mathcal{D}_{sys2}} \Big[ \lambda A \Big( \mathbf{o}_t, \pi_\theta^{off}(\mathbf{a}_t \mid \mathbf{o}_t; g_h); g_h \Big)$$
$$+ \exp\big(\beta A(\mathbf{o}_t, \mathbf{a}_t; g_h)\big) \times \log \pi_\theta^{off}(\mathbf{a}_t \mid \mathbf{o}_t; g_h) \Big] \quad (6)$$

In (6), the log-likelihood term $\log \pi_\theta^{off}(\mathbf{a}_t \mid \mathbf{o}_t; g_h)$ encourages the policy to imitate actions from the offline dataset. The exponential weight $\exp\big(\beta A(\mathbf{o}_t, \mathbf{a}_t; g_h)\big)$ modulates the strength of imitation, assigning a larger weight to actions with a higher estimated advantage. In addition, we introduce a **policy-anchored advantage term**, which regularizes the actor update by anchoring learning to actions preferred by the critic when sampling from the current policy, complementing dataset-conditioned imitation.

By incorporating critic-guided preferences, the policy-anchored update can mitigate over-imitation and reduce the tendency toward overly templated behaviors, while encouraging actions that are estimated to yield higher returns. As a result, it may improve transfer beyond the exact offline trajectories. Based on these loss functions, System 2 is fine-tuned by jointly training the actor and critic in an offline RL framework.

## 4.3. Online Training for System 2 Self-Improvement

To enable continual improvement, we introduce an additional online RL stage that retains the offline dataset while

**Algorithm 1** Multi²: Hierarchical Multi-Agent Framework

1: **Require:** $\hat{\pi}_\theta, \mathcal{D}_{sys1}, \mathcal{D}_{sys2}, \mathcal{E}$, epochs $EP_1, EP_2, EP_3$
2: Init System 1: $\pi_\phi$; System 2: $\pi_\theta^{off} \leftarrow \hat{\pi}_\theta$; critics $Q_\omega, V_\psi$; buffer $\mathcal{B}_{sys2}$
3: **for** $i = 1$ to $EP_1$ **do**
4:     Update $\pi_\phi$ on $\mathcal{D}_{sys1}$ via SFT loss (3)
5: **end for**        // System 1 training with SFT
6: **for** $j = 1$ to $EP_2$ **do**
7:     Update $Q_\omega, V_\psi, \pi_\theta^{off}$ on $\mathcal{D}_{sys2}$ via (4),(5),(6)
8: **end for**  // System 2 training with Offline RL
9: Set $\pi_\theta^{on} \leftarrow \pi_\theta^{off}, Q_\omega, V_\psi, \mathcal{B}_{sys2} \leftarrow \mathcal{D}_{sys2}$
10: **for** $k = 1$ to $EP_3$ **do**
11:     Receive task $K_n$, observation $\mathbf{o}_0$, sample sub-goal $g_h \sim \pi_\phi(\cdot \mid \mathbf{o}_0; K_n)$
12:     **for** $t = 0$ to $T - 1$ **do**
13:         Act $\mathbf{a}_t \sim \pi_\theta^{on}(\cdot \mid \mathbf{o}_t; g_h)$; step in $\mathcal{E}$ to get $(r_t, \mathbf{o}_{t+1})$
14:         Store $(g_h, \mathbf{o}_t, \mathbf{a}_t, r_t, \mathbf{o}_{t+1})$ in $\mathcal{B}_{sys2}$
15:         **if** $g_h$ achieved **then**
16:             $g_{h+1} \sim \pi_\phi(\cdot \mid \mathbf{o}_{t+1}; K_n)$
17:         **end if**
18:     **end for**
19:     Update $Q_\omega, V_\psi$ with (4), (5)
20:     Update $\pi_\theta^{on}$ with (7)
21: **end for**   // System 2 training with Online RL

incorporating newly collected interactions. This online refinement enables self-improvement by preferentially reinforcing high-return behaviors.

$$\mathcal{L}_\theta^{online} = -\mathbb{E}_{(g_h, \mathbf{o}_t, \mathbf{a}_t) \sim \mathcal{B}_{sys2}} \Big[ w_t \log \pi_\theta^{on}(\mathbf{a}_t \mid \mathbf{o}_t; g_h) \Big]$$
$$+ \eta \, \mathbb{E}_{(g_h, \mathbf{o}_t) \sim \mathcal{B}_{sys2}} \Big[ D_{KL}\big(\pi_\theta^{on} \| \pi_\theta^{off}\big)(\mathbf{o}_t; g_h) \Big] \quad (7)$$

In (7), $\mathcal{B}_{sys2}$ denotes a replay buffer that mixes the offline dataset $\mathcal{D}_{sys2}$ with trajectories newly collected by the current policy $\pi_\theta^{on}$ during online interaction. The weight $w_t$ is computed from the advantage as $w_t = \exp\big(\frac{1}{\alpha} A(\mathbf{o}_t, \mathbf{a}_t; g_h)\big)$, so that actions with higher estimated advantage are reinforced more strongly. Our key design is the **KL regularization term**, which constrains $\pi_\theta^{on}$ to stay close to the offline-trained policy $\pi_\theta^{off}$. This regularization prevents overly sharp policy shifts during online refinement, mitigates mode collapse [75, 76, 77], and stabilizes self-improvement. The coefficient $\eta$ trades off policy improvement against this stability constraint.

# 5. Experiments

We consider how to build interactive LLM-based agents that solve long-horizon tasks reliably and efficiently. Such settings are challenging because errors compound over multi-turn interaction, often causing objective drift and increased token usage. To evaluate whether Multi² addresses these

issues, we benchmark it in interactive environments that require agentic capabilities—i.e., maintaining task intent, decomposing goals, and executing valid actions under changing observations. Additional results are provided in Appendix G.

Accordingly, our experiments aim to answer three questions: (1) Does Multi² improve long-horizon task performance across diverse interactive benchmarks? (2) How token-efficient is Multi² at inference time compared to prior works? (3) Which components (role specialization, loss design, and offline-to-online training) are responsible for the gains?

## 5.1. Experiment Setups

**Benchmarks.** We evaluate our framework on Science-World [22], ALFWorld [78], and TextCraft [16], which cover complementary observation/action spaces for multi-turn interaction. ScienceWorld tests curriculum-driven, long-horizon planning in a stateful environment where actions update the world, and agents adapt to new observations. ALFWorld targets household manipulation with discrete high-level actions and sparse rewards, stressing goal maintenance and credit assignment. TextCraft evaluates recipe-based symbolic crafting, where sparse terminal rewards and compositional dependencies challenge agents to sustain intent and avoid objective drift. Additional details are in Appendix B.

**Backbones.** Our framework is built on the instruction-tuned open-source Qwen-2.5 3B [79], Mistral 7B v0.3 [80], and Llama-3.1 8B [81] models. Training is conducted on an RTX PRO 6000 Blackwell (128GB RAM), and details of the experimental setup and hyperparameters are provided in Appendix C.

**Model Architecture with Low-Rank Adaptation.** We adopt low-rank adaptation (LoRA) [82] for parameter-efficient fine-tuning, enabling role specialization without updating the full backbone. System 1 and System 2 share the same pretrained backbone but use separate LoRA adapters. During both training and inference, only the adapter corresponding to the selected system is activated.

**Datasets.** Since training System 1 and System 2 requires hierarchical datasets, we build our dataset construction pipeline on the framework introduced in Glider [15] and further modify and extend it to improve consistency and reproducibility. In particular, we refine sub-goal extraction and prompt construction with code-based rules, improving reproducibility and reducing sensitivity to design and prompting choices. As a result, the datasets are more consistent and readily reusable for training and evaluating long-horizon hierarchical agents. We release the datasets

*Table 1.* Performance comparison with baselines on long-horizon interactive environments (ScienceWorld, ALFWorld, and TextCraft) using three backbone LLMs. We report *ID/OOD* performance on ScienceWorld and ALFWorld, and overall success rate on TextCraft. Cyan highlights the best result for each backbone and evaluation setting. In Structure Type column, *Hier.* denotes a hierarchical structure.

| Base Models | Training Type | Structure Type | | Methods | ScienceWorld | | ALFWorld | | TextCraft |
| | | Single | Hier. | | ID [%] | OOD [%] | ID [%] | OOD [%] | Success [%] |
|---|---|---|---|---|---|---|---|---|---|
| Qwen-2.5 3B | Prompt-based | ✓ | | ReAct [14] | 20.82 | 9.34 | 6.72 | 5.22 | 3.00 |
| | | ✓ | | Reflexion [34] | 21.87 | 8.80 | 37.14 | 25.36 | 1.43 |
| | | | ✓ | ADaPT [16] | 20.00 | 8.53 | 15.38 | 10.26 | 21.00 |
| | Fine-tuning-based | ✓ | | GRPO [83] | 18.11 | 8.12 | 7.50 | 6.43 | 3.50 |
| | | | ✓ | Glider [15] | 54.69 | 17.75 | 52.86 | 41.43 | 18.50 |
| | | | ✓ | Multi² (Proposed) | 60.68 | 29.04 | 61.43 | 49.29 | 28.50 |
| Mistral 7B | Prompt-based | ✓ | | ReAct [14] | 23.25 | 8.91 | 4.29 | 6.72 | 23.00 |
| | | ✓ | | Reflexion [34] | 23.68 | 8.59 | 36.43 | 22.83 | 23.16 |
| | | | ✓ | ADaPT [16] | 10.50 | 1.65 | 16.67 | 5.13 | 30.20 |
| | Fine-tuning-based | ✓ | | GRPO [83] | 38.47 | 13.73 | 7.40 | 7.10 | 11.00 |
| | | | ✓ | Glider [15] | 58.33 | 28.22 | 45.00 | 45.71 | 28.50 |
| | | | ✓ | Multi² (Proposed) | 69.97 | 31.32 | 56.43 | 50.71 | 44.50 |
| Llama-3.1 8B | Prompt-based | ✓ | | ReAct [14] | 23.23 | 10.30 | 6.72 | 7.46 | 9.00 |
| | | ✓ | | Reflexion [34] | 23.20 | 6.97 | 35.71 | 29.29 | 11.00 |
| | | | ✓ | ADaPT [16] | 26.20 | 12.27 | 11.54 | 6.41 | 5.00 |
| | Fine-tuning-based | ✓ | | GRPO [83] | 25.79 | 4.61 | 8.57 | 7.50 | 5.50 |
| | | | ✓ | Glider [15] | 60.48 | 34.36 | 43.57 | 37.86 | 9.50 |
| | | | ✓ | Multi² (Proposed) | 67.61 | 30.68 | 57.86 | 56.43 | 35.60 |

and code to support broader adoption and fair comparison. Further details are provided in Appendix A.

**Metric.** We adopt a strict pass@1 metric, allowing only a single rollout without retries or resampling, to emphasize reliability and one-shot efficiency. We report pass@1 separately on *ID* and *OOD* splits defined at the task-structure level (Appendix D). *ID* tasks share goal templates and interaction patterns with those seen during training, whereas *OOD* tasks use novel templates. Importantly, evaluation is conducted in newly instantiated episodes with different object instances, layouts, and intermediate states, so even *ID* tasks are not identical to training trajectories and require state-level generalization. For prompt-based baselines, we follow standard protocols with a small set of in-context demonstrations. For fine-tuned agents, we evaluate zero-shot without in-context examples.

Reflexion [34], however, is a multi-trial method that improves through self-critique and revision across successive attempts. Accordingly, we report pass@6, consistent with its standard evaluation protocol.

### 5.2. Baselines

We compare Multi² against a deliberately selected set of baselines, including prompt-based and fine-tuning-based agents covering both single and hierarchical designs. Additional details are in Appendix E.

- **Prompt-Based (ReAct [14], Reflexion [34], ADaPT [16]):** These baselines rely on prompting without parameter updates. ReAct interleaves chain-of-thought reasoning and actions, Reflexion adds self-reflection memory across subsequent attempts, and ADaPT uses a planner–executor hierarchy for sub-goal generation and execution.

- **Fine-Tuning-Based (GRPO [83], Glider [15]):** These baselines update model parameters. GRPO is a single-agent RL method that directly optimizes a unified policy. Glider adopts a hierarchical framework with a high-level planner and a low-level controller. However, during online adaptation, it mainly updates the high-level planner while keeping the low-level executor largely fixed, limiting its ability to correct execution-side failures from environment feedback. In contrast, Multi² explicitly decouples the adaptation of different functional roles, rather than relying on a single shared LoRA adapter and prompt-based role separation as in Glider.

- **Multi² (Proposed):** It considers a hierarchical structure that separates System 1 for high-level planning and System 2 for action execution. System 1 is trained with SFT, while System 2 is optimized through offline-to-online RL for role specialization.

### 5.3. Performance Comparison

Table 1 summarizes overall results on interactive environments. Notably, Multi² achieves the strongest performance across most backbones and splits, suggesting improved generalization across model settings. Prompt-based baselines (e.g., ReAct and Reflexion) perform poorly, which is consistent with compounding errors and drift in long-horizon

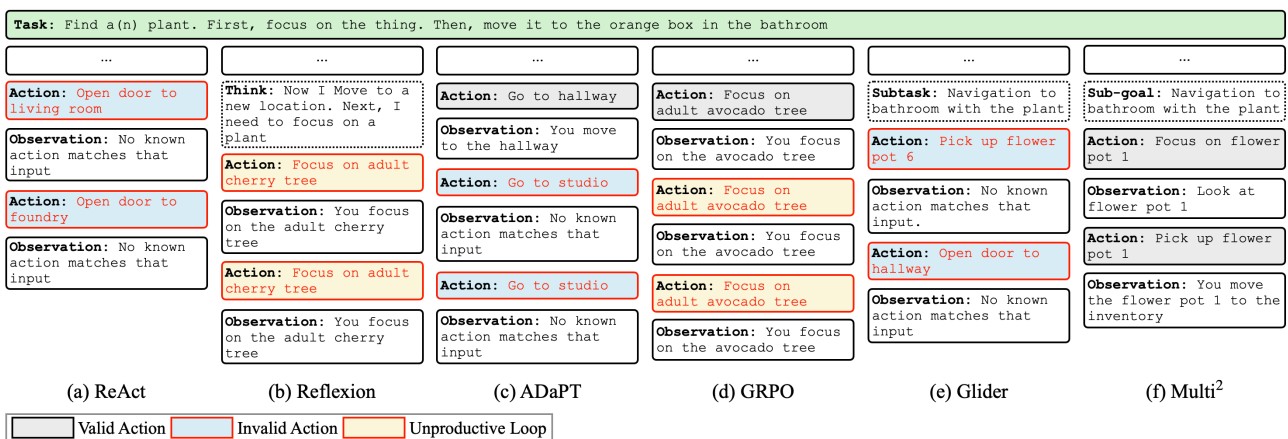

*Figure 3.* Representative failure examples of baseline agents on the ScienceWorld `Find-a-plant` task with Qwen-2.5 3B. Each column corresponds to a different method (a–f). Shaded boxes indicate action types: gray denotes valid actions, blue denotes invalid actions rejected by the environment, and yellow denotes unproductive loops that repeat actions without measurable progress.

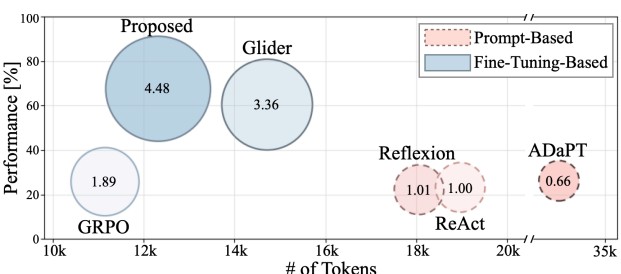

*Figure 4.* Inference-time token efficiency on ScienceWorld *ID* split with Llama-3.1 8B. The $x$-axis denotes token usage, and the $y$-axis denotes performance. Bubble size indicates token efficiency, computed as performance/tokens and normalized such that ReAct is 1.0. Solid lines: fine-tuning-based; dashed lines: prompt-based.

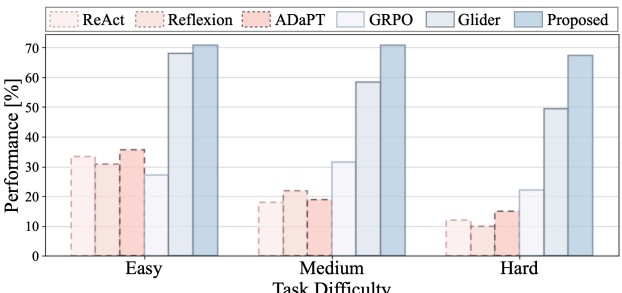

*Figure 5.* Objective drift robustness on the ScienceWorld *ID* split with Llama-3.1 8B, stratified by task difficulty. Bars report performance grouped by task difficulty, where harder tasks typically require longer multi-turn interaction and sustained goal tracking.

tasks. Here, ADaPT outperforms GRPO on most tasks because its hierarchical prompting provides a useful direction. Among fine-tuning-based baselines, Glider improves over prompt-based methods, indicating that hierarchical structure and parameter updates help mitigate long-horizon brittleness. However, Glider relies primarily on prompt-based role separation while sharing a single adapter across the planner and executor, which can limit role specialization compared to a fully decoupled design. In contrast, Multi² addresses this challenge by decoupling System 1 and System 2 with role-specialized training, and by optimizing System 2 using the proposed RL pipeline and well-structured loss function.

## 5.4. Qualitative Error Analysis

When an agent's plan drifts from the intended objective, its actions can become misaligned with the environment, often triggering spirals of invalid commands. Figure 3 shows representative failures on the ScienceWorld `Find-a-plant` task, where objective drift appears as invalid actions and unproductive loops. In Figure 3(f), Multi² mitigates objec-

tive drift by anchoring execution to explicit sub-goals and improving action validity via RL fine-tuning. In contrast, prompt-based agents often persist in rejected commands or repeat actions (Figures 3(a–c)), indicating limited recovery once trajectories deviate. Fine-tuning-based agents likewise tend to commit to an implausible target and continue with non-progressing or unsupported actions (Figures 3(d–e)). Overall, these comparisons suggest that Multi² better maintains goal-consistent execution under multi-turn interaction, reducing both constraint violations and stalled behavior.

## 5.5. Token Efficiency Analysis

Figure 4 plots token efficiency related to inference-time token usage. Multi² is consistently more token-efficient, benefiting from hierarchical, on-demand invocation that avoids unnecessary interaction and from RL-based fine-tuning that produces well-formed actions without long prompts. In contrast, prompt-based methods rely on in-context demonstrations, and even strong fine-tuned baselines typically require more tokens to achieve comparable performance. Overall, Multi² attains better results with fewer inference tokens.

*Table 2.* Training assignments of ablation study: the learning paradigm (SFT and RL) applied to `System 1` and `System 2` under different configurations.

|  | (1) RL-SFT | (2) Only RL | (3) Only SFT | (4) Proposed |
|---|---|---|---|---|
| `System 1` | RL | RL | SFT | SFT |
| `System 2` | SFT | RL | SFT | RL |

*Table 3.* Comparison of structure, parameterization, and decision flow for the ablations. In the Decision Flow column, *Inform.* denotes the task description and current observation, which serve as inputs to `System 1`.

|  | Hierarchy | Adapter | Decision Flow | Output |
|---|---|---|---|---|
| **Single** | ✗ | Shared | Inform. | Action |
| **Shared** | ✓ | Shared | Inform. → Sub-goal | Action |
| **Proposed** | ✓ | Role-Specific | Inform. → Sub-goal | Action |

## 5.6. Objective Drift Robustness Analysis

Figure 5 reports performance across task difficulty (Easy/Medium/Hard) (Appendix D), where harder tasks require longer multi-turn interaction and sustained goal tracking, increasing vulnerability to objective drift. Prompt-based agents degrade sharply as difficulty increases, suggesting limited ability to preserve intent and recover from errors over long horizons. Fine-tuning-based agents substantially improve performance, yet show a noticeable drop from medium to hard, indicating that decomposition alone does not fully resolve long-horizon brittleness. In contrast, `Multi²` remains consistently strong, thereby widening the gap on harder tasks and underscoring the importance of mitigating objective drift for reliable agentic behavior.

## 5.7. Ablation Studies

To understand the contribution of each component in our framework, we conduct ablation studies across training configurations, model structure, adapter design, objective functions, and model scales on the ScienceWorld benchmark. These analyses isolate the effects of role specialization, hierarchical decomposition, and our offline-to-online training objective, allowing us to assess how each design choice contributes to stable and effective long-horizon interaction.

**Training Configurations.** Table 2 summarizes the training assignments for `System 1` and `System 2`, and Figure 6(a) shows the resulting performance distributions. Overall, the ablations support our role specialization: SFT yields structured, context-aware planning in `System 1`, while RL improves action-level robustness in `System 2`. The swapped setup **RL-SFT** performs worst, indicating a clear role mismatch. RL is unstable for context-aware task decomposition, whereas an SFT-trained `System 2` is brittle under subtle environment changes. **Only RL** shows a lower median and substantially larger variance, suggesting that RL-only train-

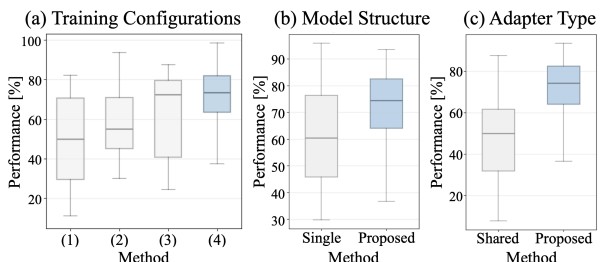

*Figure 6.* Ablation results on the ScienceWorld *ID* split using Llama-3.1 8B. (a) **Training Configurations**: (1) RL-SFT, (2) only RL, (3) only SFT, and (4) the proposed role-specialized setting, where `System 1` is trained with SFT and `System 2` is trained with RL. (b) **Model Structure**: Single model vs. the proposed hierarchical structure. (c) **Adapter Type**: Shared adapter vs. the proposed role-specific adapters.

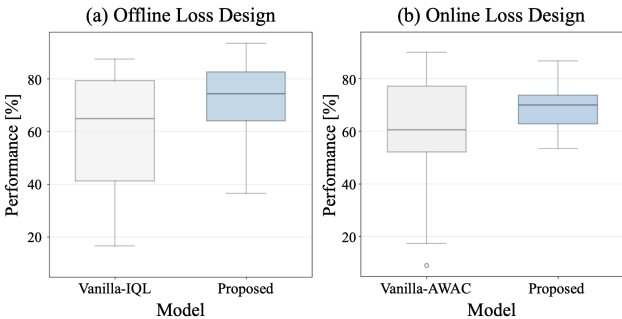

*Figure 7.* Ablation results on the ScienceWorld *ID* split using Llama-3.1 8B (a) and Qwen-2.5 3B (b). (a) **Offline Loss Design**: Vanilla-IQL (without policy-anchored advantage term) vs. the proposed offline objective. (b) **Online Loss Design**: Vanilla-AWAC (without KL regularization) vs. the proposed online objective.

ing is unstable for high-level planning. **Only SFT** achieves competitive median performance but is less robust, with frequent failures. These results confirm that the proposed training pipeline is critical for robust long-horizon behavior.

**Model Structure.** Table 3 summarizes the structural, parameter-level, and information-flow differences between the compared variants. Figure 6(b) compares a flat **Single** model with the proposed hierarchical design on the Science-World *ID* split. The hierarchical variant achieves a clearly higher median performance and an overall upward-shifted distribution relative to the single model baseline, suggesting that the gain is not explained solely by the RL training pipeline. Instead, explicit hierarchical decomposition itself contributes meaningfully to performance.

**Adapter Type.** Figure 6(c) represents the effect of role-specific adapters . Compared with the **Shared** adapter variant, the proposed design yields a markedly higher median and an upward-shifted interquartile range, indicating that parameter decoupling improves not only peak performance but also the overall performance distribution. These results

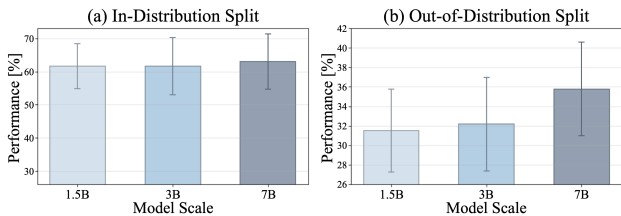

*Figure 8.* Effect of backbone model scale on Multi² performance in ScienceWorld with Qwen-2.5 backbones at three scales (1.5B, 3B, and 7B). (a) **In-Distribution Split** and (b) **Out-of-Distribution Split**. Error bars denote one standard deviation over runs.

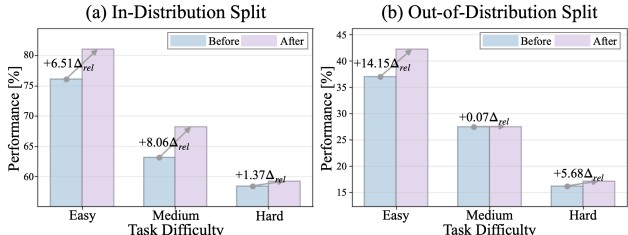

*Figure 9.* Online RL adaptation on ScienceWorld with Qwen-2.5 3B. We report performance after online RL compared to offline trained policy, grouped by task difficulty (Easy/Medium/Hard) on (a) **In-Distribution Split** and (b) **Out-of-Distribution Split**.

support the importance of separate adapters for effective role specialization of System 1 and System 2.

**Loss Function Designs.** Figure 7 compares the System 2 objective against vanilla variants: (a) **Vanilla-IQL** without the policy-anchored advantage term and (b) **Vanilla-AWAC** without KL regularization during online adaptation. In (a), our offline objective raises the median and shrinks the low-performance tail, suggesting that the policy-anchored term improves policy refinement beyond conservative dataset imitation alone. In (b), Vanilla-AWAC occasionally performs well but exhibits larger variance and more pronounced low-end failures, whereas our online objective produces a tighter and more reliable performance distribution. Taken together, these results suggest that the gains are not merely due to stronger optimization, but to an objective design that explicitly stabilizes execution in long-horizon interaction—offline through critic-guided policy anchoring, and online through KL-regularized refinement.

**Model Scales.** To examine the effect of model capacity, we run Multi² with Qwen-2.5 models at three scales (1.5B, 3B, and 7B) and evaluate on *ID* and *OOD* splits (Figure 8). On *ID* split, performance is largely comparable across model sizes, indicating limited sensitivity to capacity within this range. In contrast, on *OOD* split, performance increases monotonically with scale, suggesting that larger backbones better handle distribution shift within our pipeline.

**5.8. Online Adaptation Analysis**

Figure 9 compares the effect of online RL over an offline-trained policy across task difficulty. On the *ID* split, online RL yields consistent improvements on easy and medium tasks, while maintaining comparable performance on hard tasks, suggesting that online updates primarily refine execution and reduce compounding errors without destabilizing in-distribution behavior. On the *OOD* split, the gains from online adaptation become more pronounced. We attribute this larger benefit to the ability of online interaction to correct execution-level errors under distribution shift, where mismatches between sub-goals and environment dynamics

are more frequent. Overall, online RL improves the reliability of execution in multi-turn interaction. Additional analyses in the online training are described in Appendix G.

## 6. Conclusion

We propose Multi², a hierarchical multi-agent framework for long-horizon interactive environments, explicitly addressing two practical bottlenecks of LLM-based agents: objective drift and poor token efficiency. Our key contribution is a principled decoupling of planning and execution, where System 1 generates context-aware sub-goals while System 2 executes atomic actions conditioned on them, enabling stable coordination over long horizons. To support this structure, we introduce a unified training pipeline that combines policy-anchored offline RL with KL-regularized online refinement, providing a stable mechanism for continual improvement under multi-turn interaction. Across diverse interactive benchmarks, Multi² consistently improves performance while yielding better token efficiency and stronger robustness to objective drift. We also release our processed hierarchical datasets, enabling broader adoption in the community.

## Impact Statement

This paper advances methods for agentic AI–LLM-based agents that can reliably and efficiently interact with dynamic environments over long horizons. By contributing to more effective and principled learning algorithms, this work has the potential to benefit a wide range of applications and, when deployed responsibly, improve productivity and quality of life. As with most general-purpose machine learning techniques, the proposed methods could be applied in both beneficial and potentially harmful contexts depending on the application domain and deployment practices. We do not identify ethical concerns or negative societal consequences unique to this work beyond those commonly associated with broader deployment of capable interactive AI systems. Responsible use of the proposed methods should follow established best practices regarding safety, fairness, transparency, and accountability.

# Acknowledgements

We thank the ICML 2026 reviewers for their constructive feedback. This work was supported by the Institute of Information & Communications Technology Planning & Evaluation (IITP) grant funded by the Korea government (MSIT) (No. RS-2026-25519475), the National Research Foundation of Korea (NRF) grant funded by the Korean government (MSIT) under Grant RS-2025-02214082, the NRF grant funded by the Korea government (MSIT) (RS-2023-00278812), and Korea Institute of Police Technology (KIPoT) funded by the Korean National Police Agency & Korea Customs Service (RS-2026-25536784).

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

# Appendix

# A. Dataset Construction

We describe the dataset construction pipeline for the role-specific hierarchical datasets. Building on `Glider` [15], we extend the processing to better match our hierarchical setting. Concretely, we generate the dataset programmatically via a code-based pipeline that loads GPT-4-Turbo conversation trajectories released by AgentGym [84] and converts them into role-specific training instances. We parse these trajectories into supervision signals aligned with high-level planning and low-level execution.

We make two modifications to improve stability and reproducibility: deterministic, rule-based sub-goal extraction (e.g., step segmentation, command normalization, and filtering out non-actionable text) and standardized prompt templates per role with fixed instruction headers and output schemas to reduce prompt sensitivity. Finally, we convert each episode into (1) observation-sub-goal pairs for `System 1` and (2) sub-goal-conditioned low-level transitions for `System 2`.

## A.1. Dataset Examples

```
{
  "Task Description": "Your task is to boil ice cream. For compounds without a boiling point,
combusting the substance is also acceptable. First, focus on the substance. Then, take actions that
will cause it to change its state of matter.",
  "Sub-goal": [
    "Navigate to the kitchen",
    "Collect tools: thermometer and metal pot",
        …
  ],
  "Action": [
    "open door to kitchen",
    "go to kitchen",
    "look around",
    …
    "wait"
  ],
  "Obs": [
    "This room is called the hallway. In it, you see: \n\tthe agent\n\ta substance called air\n\ta
picture\nYou also see:\n\tA door to the art studio (that is closed)\n\tA door to the bedroom (that
is closed)\n\tA door to the greenhouse (that is closed)\n\tA door to the kitchen (that is closed)
\n\tA door to the living room (that is closed)\n\tA door to the workshop (that is closed)"
    "The door is now open.",
    "You move to the kitchen.",
    …
    "You decide to wait for 1 iterations."
  ],
  "Next Obs": [
    "The door is now open.",
    "You move to the kitchen.",
    "This room is called the kitchen. In it, you see:\n\tthe agent\n\ta substance called air\n\ta
cupboard (closed)\n\ta freezer (closed)\n\ta stove (off)\n\ta thermometer (10°C)\n\ta glass jar
(containing salt)",
    …
    "You decide to wait for 1 iterations."
  ],
  "Reward": [
    0.0, 0.0, 0.0, …, 1.0
  ],
  "Score": [
    0.0, 0.0, 0.0, …, 1.0
  ],
  "Done": [
    false, false, false, …, true
  ]
}
```

*Figure 10.* Training dataset example for `System 1` and `System 2` on ScienceWorld benchmark.

```
{
  "Task Description": "Your task is to examine the book with the desklamp",
  "Sub-goal": [
    "Retrieve book (It is likely on bed, but maybe elsewhere)",
    "Use desklamp"
  ],
  "Action": [
    "Go to bed 1",
    "Take book 1 from bed 1",
    "Go to desk 1",
    "Go to dresser 1",
    …
    "Use desklamp 1"
  ],
  "Obs": [
    "You are in the middle of a room. Looking quickly around you, you see a bed 1, a desk 1, a
drawer 5, a drawer 4, a drawer 3, a drawer 2, a drawer 1, a dresser 1, a garbagecan 1, a
laundryhamper 1, and a shelf 1.",
    "You arrive at bed 1. On the bed 1, you see a book 1, a laptop 1, and a pillow 1.",
    "You pick up the book 1 from the bed 1.",
    "You are in the middle of a room. Looking quickly around you, you see a bed 1, a desk 1, a
drawer 5, a drawer 4, a drawer 3, a drawer 2, a drawer 1, a dresser 1, a garbagecan 1, a
laundryhamper 1, and a shelf 1. You pick up the book 1 from the bed 1.",
    …
    "You won!"
  ],
  "Next Obs": [
    "You arrive at bed 1. On the bed 1, you see a book 1, a laptop 1, and a pillow 1.",
    "You pick up the book 1 from the bed 1.",
    "You are in the middle of a room. Looking quickly around you, you see a bed 1, a desk 1, a
drawer 5, a drawer 4, a drawer 3, a drawer 2, a drawer 1, a dresser 1, a garbagecan 1, a
laundryhamper 1, and a shelf 1. You pick up the book 1 from the bed 1.",
    "You arrive at desk 1. On the desk 1, you see an alarm clock 1, a bowl 1, a CD 2, a mug 3, a
mug 2, a mug 1, a pen 1, and a pencil 2.",
    …
    "You won!"
  ],
  "Reward": [
    0.0, 0.0, 0.0, …, 1.0
  ],
  "Score": [
    0.0, 0.0, 0.0, …, 1.0
  ],
  "Done": [
    false, false, false, …, true
  ]
}
```

*Figure 11.* Training dataset example for System 1 and System 2 on ALFWorld benchmark.

```
{
  "Task Description": "Craft green stained glass\n\nCrafting commands:\ncraft 8 green carpet using
8 white carpet, 1 green dye\ncraft 8 green stained glass pane using 8 glass pane, 1 green
dye\ncraft 8 lime stained glass using 8 glass, 1 lime dye\ncraft 8 green stained glass using 8
glass, 1 green dye\ncraft 8 green terracotta using 8 terracotta, 1 green dye\ncraft 8 white stained
glass using 8 glass, 1 white dye\ncraft 8 orange stained glass using 8 glass, 1 orange dye\ncraft 8
brown stained glass using 8 glass, 1 brown dye\n",
  "Sub-goal": [
    "Acquire needed items (may need: 8 glass, 1 green dye) for green stained glass",
    "Craft target: green stained glass"
  ],
  "Action": [
    "Craft 8 green stained glass using 8 glass, 1 green dye",
    "Get 8 glass",
    "Get 1 green dye",
    ...
    "Craft 8 green stained glass using 8 glass, 1 green dye"
  ],
  "Obs": [
    "(start)",
    "Could not find enough items to craft minecraft:green_stained_glass",
    "Got 8 glass",
    "Got 1 green dye",
    ...
    "Task completed"
  ],
  "Next Obs": [
    "Could not find enough items to craft minecraft:green_stained_glass",
    "Got 8 glass",
    "Got 1 green dye",
    ...
    "Task completed"
  ],
  "Reward": [
    0.0, 0.0, ..., 1.0
  ],
  "Score": [
    0.0, 0.0, ..., 1.0
  ],
  "Done": [
    false, false, ..., true
  ]
}
```

*Figure 12.* Training dataset example for `System 1` and `System 2` on TextCraft benchmark.

Figures 10, 11, and 12 show training dataset examples for each benchmark (ScienceWorld, ALFWorld, and TextCraft). We formulate all environments as partially observable Markov decision processes and use separate prompt templates for `System 1` and `System 2` to match their input–output requirements. Each `System 1` instance in $\mathcal{D}_{sys1}$ pairs the task description and observation with the extracted sub-goal. Each `System 2` instance in $\mathcal{D}_{sys2}$ is a sub-goal-conditioned transition (e.g., $(\mathbf{o}_t, \mathbf{a}_t, \mathbf{o}_{t+1})$) with trajectory metadata (reward/score/done).

## A.2. Role-Specific Prompts

We use separate prompts to reflect the distinct input-output requirements of the planner (System 1) and executor (System 2): System 1 maps {task, observation} to a sub-goal, while System 2 maps {sub-goal, observation} to an executable action and a completion signal.

---

**System 1 Prompt.**

You are a high-level planner. Based on the state (task description and current observation), please generate a clear and simple sub-goal. \n
**Task description:** \n
**Grouped action:** []. **Current observation:** \n

---

**System 2 Prompt.**

You are a low-level action executor. Based on the current sub-goal and observation, please generate an executable action and determine whether the sub-goal has been completed (true/false). \n
**Sub-goal:** \n
**Observation:** \n

---

### A.3. Reward Shaping

In ScienceWorld [22], the environment provides a dense progress reward in $[0, 1]$ (along with a success flag) reflecting partial completion of the underlying sub-goals; we follow the official reward definition and directly use the environment-provided reward signal.

In ALFWorld [78], the environment reward is sparse: the agent receives a task-level reward of $1$ only upon completing the full task, and $0$ otherwise. To enable hierarchical training, we introduce an internal sub-goal-level shaping reward used only for training System 2. Given the extracted sub-goal sequence, we segment each trajectory into contiguous intervals where one sub-goal is active, and assign an internal reward of $1$ when the active sub-goal is completed (as detected by our trajectory parser), and $0$ otherwise. Importantly, the environment reward remains unchanged: the task-level reward is provided only when the full task succeeds.

In TextCraft [16], the environment is also sparse: the agent receives a reward of $1$ only upon successfully crafting the target item (and $0$ otherwise), under a constrained command interface (e.g., craft/get/inventory). We therefore apply the same hierarchical shaping strategy, awarding intermediate reward when the currently active sub-goal (e.g., the next recipe step/prerequisite) is satisfied, while preserving the original terminal reward for task completion.

### A.4. Reproducibility and Release

All sub-goal extraction rules, prompt templates, and preprocessing scripts are implemented as deterministic, code-based transformations, minimizing sensitivity to manual design choices and ensuring that the resulting datasets are consistent and reusable. We publicly release the processed role-specific datasets, the exact prompt templates and extraction rules, as well as the training and evaluation code at https://anonymous-projectpage.github.io/Multi-Square/. This release enables reproducible training and transparent evaluation, supporting fair comparison across long-horizon LLM agent methods.

## B. Benchmarks

We evaluate long-horizon decision-making on three benchmarks that differ in dynamics, action constraints, and feedback signals. ScienceWorld [22] provides a score considering incremental progress, whereas ALFWorld [78] and TextCraft [16] are primarily evaluated by success rate.

### B.1. ScienceWorld

**Overview.** ScienceWorld [22] is a text-only interactive world aimed at measuring scientific reasoning in the context of simple, experiment-like procedures. The benchmark consists of 30 tasks drawn from 10 topical areas and includes comparatively rich simulated phenomena (e.g., circuits, phase changes, chemistry, and biology), making it more stateful than many standard text-game settings.

**Task Example.** Solving a task usually requires carrying out multiple steps with appropriate tools—for instance, assembling a circuit to check conductivity, inducing and verifying a phase transition (melting/boiling), or following a sequence of lab-style operations.

---

**ScienceWorld Task Example.**

**Task:** Turn on the green light bulb. First, focus on the green light bulb. Then, create an electrical circuit that powers it on.

---

**Observation.** At each step, the agent receives a natural-language description that includes the current location, visible objects (often with attributes), and outcome feedback from the previous action (e.g., inventory changes or success/failure messages). Because relevant information only becomes available after navigation and interaction, the environment is effectively partially observable.

**Action.** The agent issues executable textual commands grounded in entities (e.g., `focus on`, `pick up`, `open/close`, `connect`, `move ... to ...`). ScienceWorld exposes 25 abstract action templates that are instantiated with object arguments, and it supports a valid-action checking mechanism that can enumerate which commands are admissible under the current state constraints.

**Reward.** Rather than a purely terminal reward, ScienceWorld returns a scalar `SCORE` that increases as the agent makes progress and reaches 1.0 upon completing the task. We report ScienceWorld results using this score as the performance metric.

### B.2. ALFWorld

**Overview.** ALFWorld [78] is a text-only interactive benchmark derived from the embodied ALFRED household domain, implemented on top of TextWorld. It targets long-horizon household tasks that require both navigation and object manipulation under textual observations and action commands.

**Task Example.** The benchmark spans six high-level task categories, including pick-and-place objectives and multi-step routines such as cleaning, heating, or cooling an object before placing it at a target location (e.g., "place a clean mug on a desk").

---

**ALFWorld Task Example.**

**Task:** Clean some spatula and put it in diningtable.

---

**Observation.** At each step, the agent receives a natural-language description of the current room and the set of visible objects and receptacles, together with the goal instruction. Typically, the agent begins with a list of room contents and must then navigate across locations, open containers, and manipulate objects to satisfy the goal.

**Action.** The action space consists of template-based text commands covering navigation and manipulation, such as `go to <receptacle>` and `take <obj> from <receptacle>`, as well as task-specific interactions (e.g., `use <tool>`). For the text-game setting, the environment can also provide the set of currently admissible commands (e.g., `info['admissible_commands']`), making action validity constraints explicit.

**Reward.** ALFWorld provides sparse feedback, where success is determined by completing the full instruction. We report results using task success (success rate), following standard evaluation practice.

### B.3. TextCraft

**Overview.** TextCraft [16] is a text-based crafting benchmark proposed by ADaPT [16], motivated by Minecraft-style recipe systems. Compared to navigation-centric environments, TextCraft primarily stresses compositional goal decomposition: achieving the final target typically requires crafting a sequence of prerequisite items recursively.

**Task Example.** A typical instance asks the agent to produce a high-level item (e.g., `beehive`) that depends on intermediate ingredients (e.g., `honeycomb` and `planks`), which may themselves require multiple lower-level crafting steps.

---

**TextCraft Task Example.**

**Task:** Craft polished blackstone wall.
**Crafting commands:**
- craft 6 blackstone slab using 3 blackstone
- craft 4 polished blackstone stairs using 6 polished blackstone
- craft 1 polished blackstone pressure plate using 2 polished blackstone
- craft 6 blackstone wall using 6 blackstone
- craft 4 polished blackstone using 4 blackstone
- craft 6 polished blackstone wall using 6 polished blackstone
- craft 1 polished blackstone button using 1 polished blackstone
- craft 4 polished blackstone bricks using 4 polished blackstone
- craft 4 blackstone stairs using 6 blackstone
- craft 6 polished blackstone slab using 3 polished blackstone

---

**Observation.** The environment provides the target item together with the current crafting context, including recipe information and textual feedback from executed actions. Since state transitions are deterministic, the agent can track progress via explicit messages (e.g., `Got ...` or `Crafted ...`) and by maintaining an internal inventory state.

**Action.** The interaction space consists of commands for gathering resources and applying recipes, such

as get `<item>` and structured crafting instructions (e.g., `craft 4 oak planks using 1 oak log`). In some cases, the environment permits category-level references (e.g., requesting "planks"), requiring the agent to resolve the request into a valid concrete item (e.g., `oak planks`).

**Reward.** TextCraft is scored by whether the agent completes the target objective, and we report the success rate accordingly. The benchmark includes tasks with recipe dependency depth ranging from 2 to 4, and the official evaluation includes a 200-task test suite spanning these depths.

### B.4. Qualitative Comparison

The three benchmarks probe complementary abilities required for long-horizon agents.

- **ScienceWorld** [22]: This benchmark tests stateful interaction under rich dynamics, requiring agents to reason about changing object states (e.g., heating, mixing, connecting) and to issue valid, grounded commands from a large, constraint-driven action space, while receiving shaped score feedback.

- **ALFWorld** [78]: This benchmark emphasizes goal-directed household execution; agents must navigate across rooms and manipulate objects, yet evaluation is primarily by final task success, making recovery from small mistakes critical. Moreover, many objects have indexed identifiers, so selecting the wrong instance can cause failure even when the action is semantically correct.

- **TextCraft** [16]: This benchmark focuses on *compositional planning*; although interaction templates are relatively simple, agents must recursively decompose a target recipe into prerequisite sub-recipes (depth up to 4) and accurately track ingredients, stressing long-horizon planning and progress bookkeeping.

### B.5. Out-of-Scope: Static Math and Code Generation Benchmarks

Our study targets long-horizon decision-making, where an agent repeatedly interprets partial observations, selects an atomic action under environment constraints, and receives feedback through state transitions and rewards. This setting is central to the objective drift and robustness issues we analyze, as errors can compound over multi-turn interaction and invalid actions can stall progress. [22, 78, 16]

**Why Math and Coding Benchmarks are Not a Good Fit.** While math and code-generation benchmarks are valuable for evaluating reasoning and synthesis, they are not well aligned with our goal of assessing *agentic* behavior in interactive environments. Our setting centers on a closed-loop state-action process, where each action changes the environment and the agent must continually re-ground its decisions in new observations over multiple steps. In contrast, most math/coding benchmarks primarily evaluate producing a correct final output under a static specification, so they do not stress key failure modes in interactive agents such as constraint-violating actions, compounding execution errors, and recovery from partial progress. Accordingly, we focus on interactive benchmarks that explicitly test long-horizon decision-making under evolving world states.

**Scope Statement.** Accordingly, we restrict evaluation to interactive environments that explicitly instantiate the observation-action-feedback loop and expose long-horizon failures such as compounding errors, invalid-action spirals, and objective drift.

## C. Model Structure

### C.1. Input-Output Examples

**ScienceWorld.** We show an example of `Multi²` interacting with ScienceWorld. `System 1` generates a sub-goal from the task description and the current state summary (`group_action` + observation), and `System 2` executes low-level actions until the sub-goal is marked done.

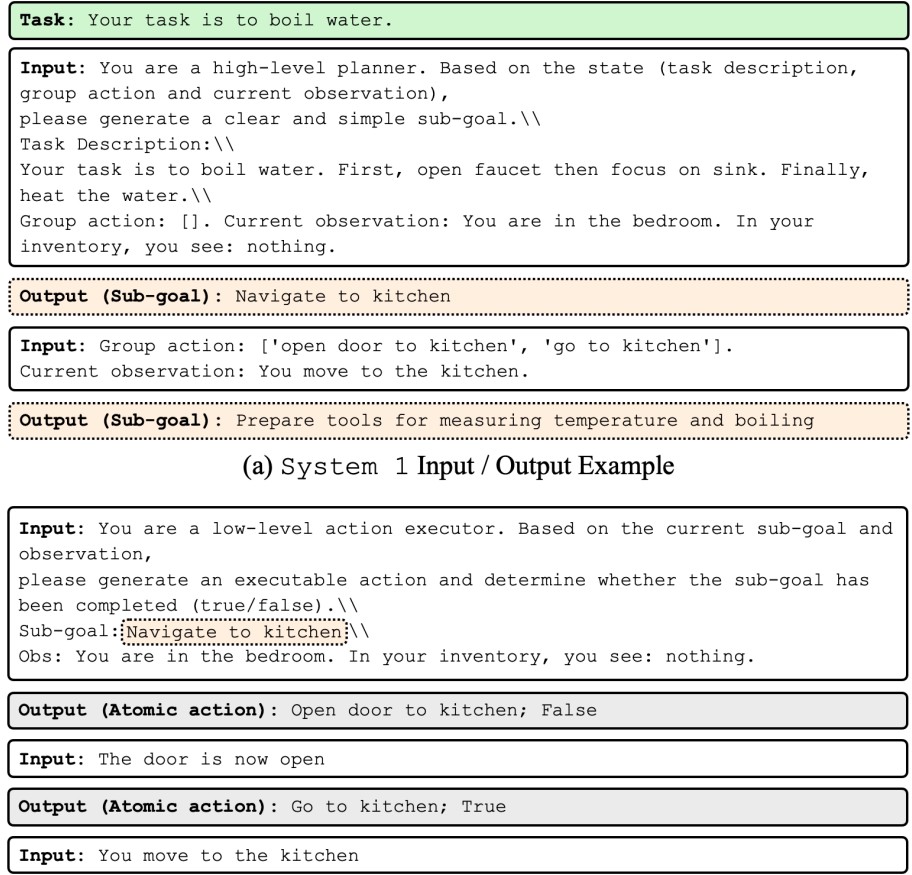

(a) `System 1` Input / Output Example

(b) `System 2` Input / Output Example

*Figure 13.* Input-output examples of `Multi²` on ScienceWorld. (a) `System 1` takes the task description and the current state summary (`group_action` and observation) and generates a sub-goal. (b) `System 2` conditions on the sub-goal and the current observation to output an atomic action and a completion flag, repeating until the sub-goal is marked done.

**System 1 (planner).** In the shown episode (task: `boil water`) in Figure 13(a), the initial state places the agent in the bedroom. Since ScienceWorld is partially observable, the agent cannot directly know the states of unvisited locations. Therefore, `System 1` proposes the sub-goal `navigate to kitchen` to gather the necessary information and access task-relevant objects.

**System 2 (executor).** Conditioned on the sub-goal and the current observation, `System 2` outputs an executable atomic action along with a boolean completion flag and repeats until the sub-goal is completed. In Figure 13(b), `System 2` emits actions such as `open door to kitchen` and `go to kitchen`, and sets the completion flag once the agent reaches the target location, at which point control returns to `System 1`.

**ALFWorld.** ALFWorld provides partially observed, text-based observations describing the current room state (visible objects/containers) and requires actions to follow a constrained command interface. We summarize interaction history with `group_action` (a running list of executed commands), which is concatenated with the current observation as a compact state summary.

```
Task: Your task is to put a creditcard in shelf.
```

```
Input: You are a high-level planner. Based on the state (task description,
group action and current observation),
please generate a clear and simple sub-goal.\\
Task Description:\\
Your task is to put a creditcard in shelf.\\
Group action: []. Current observation: You are in the middle of a room.
Looking quickly around you, you see a bed 1, a desk 1, a drawer 2, a drawer
1, a garbagecan 1, a shelf 1, and a sidetable 1.
```

```
Output (Sub-goal): Retrieve creditcard (it is likely on drawer, but maybe
elsewhere)
```

```
Input: Group action: ['go to sidetable 1', 'go to shelf 1', 'go to drawer 1',
'go to drawer 2', 'open drawer 2', 'take creditcard 2 from drawer 2'].
Current observation: You pick up the creditcard 2 from the drawer 2.
```

```
Output (Sub-goal): Move creditcard to shelf
```

(a) `System 1` Input / Output Example

```
Input: You are a low-level action executor. Based on the current sub-goal and
observation,
please generate an executable action and determine whether the sub-goal has
been completed (true/false).\\
Sub-goal: Retrieve creditcard (it is likely on drawer, but maybe elsewhere)
Obs: You are in the middle of a room. Looking quickly around you, you see a
bed 1, a desk 1, a drawer 2, a drawer 1, a garbagecan 1, a shelf 1, and a
sidetable 1.
```

```
Output (Atomic action): Go to sidetable 1; False
```

```
Input: You arrive at sidetable 1. On the sidetable 1, you see a book 2, a
keychain 1, and a pencil 3.
```

```
Output (Atomic action): Go to shelf 1; False
```

```
Input: You arrive at shelf 1. On the shelf 1, you see a pen 1, and a pencil 1.
```

(b) `System 2` Input / Output Example

*Figure 14.* Input-output examples of `Multi²` on ALFWorld. (a) `System 1` takes the task description and the current state summary (`group_action` and observation) and generates a sub-goal. (b) `System 2` conditions on the sub-goal and the current observation to output an atomic action and a completion flag, repeating until the sub-goal is marked done.

**System 1 (planner).** As shown in Figure 14(a), `System 1` takes the task description together with the state summary (`group_action` + current observation) and outputs a high-level sub-goal. In the example task (*"put a creditcard in shelf"*), the initial observation lists multiple candidate receptacles (e.g., drawers, shelf, sidetable), so `System 1` first proposes the sub-goal `Retrieve creditcard`.

**System 2 (executor).** Figure 14(b) shows `System 2`'s role as a low-level action executor. Given the current sub-goal and the current observation, `System 2` outputs an executable atomic command (e.g., `Go to sidetable 1`, `Go to shelf 1`) and a boolean completion flag. `System 2` iteratively emits commands with done=`False` while progressing toward the sub-goal, and sets done=`True` once the sub-goal condition is satisfied, at which point control returns to `System 1` for the next sub-goal.

**TextCraft.** TextCraft is a recipe-driven crafting environment: tasks specify the target item and available recipes, and actions follow a constrained command interface (e.g., `get`/`craft` with quantities).

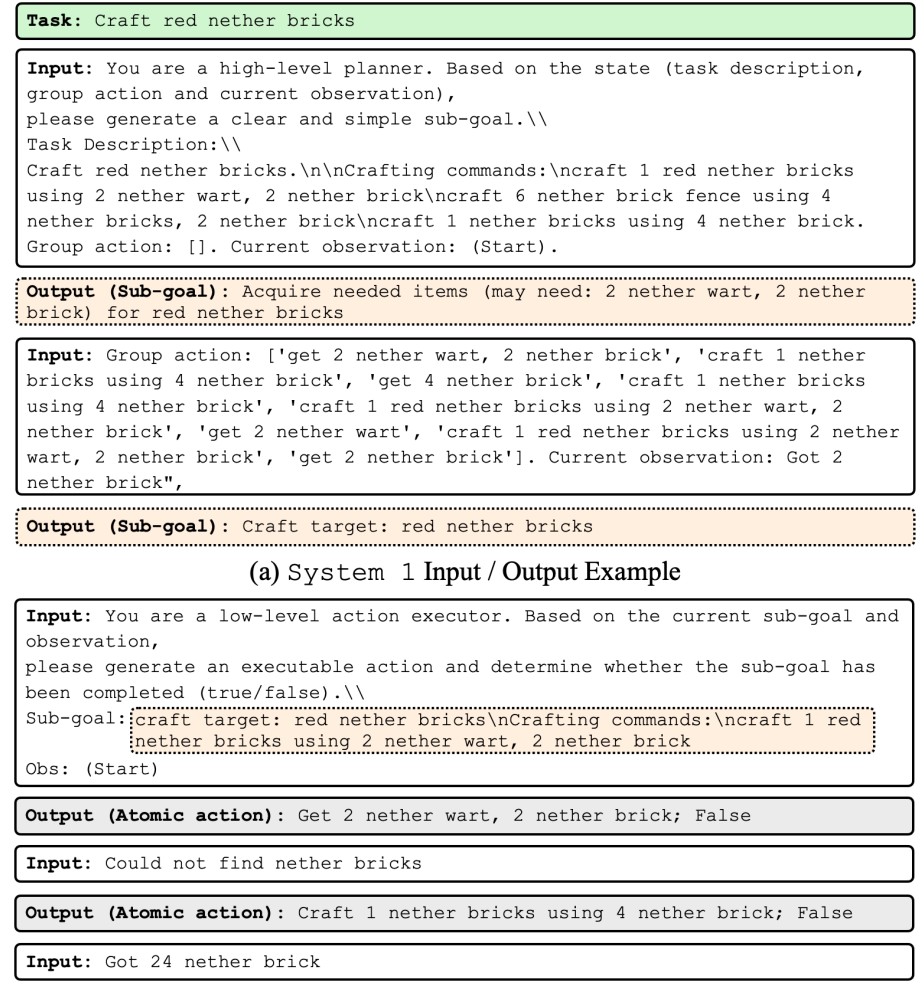

(a) `System 1` Input / Output Example

(b) `System 2` Input / Output Example

*Figure 15.* Input-output examples of `Multi²` on TextCraft. (a) `System 1` takes the task description and the current state summary (`group_action` and observation) and generates a sub-goal. (b) `System 2` conditions on the sub-goal and the current observation to output an atomic action and a completion flag, repeating until the sub-goal is marked done.

**System 1 (planner).** As shown in Figure 15(a), `System 1` proposes a prerequisite-oriented sub-goal such as `Acquire needed items (2 nether wart, 2 nether brick)` in the example task (`Craft red nether bricks`), and after the history indicates that the required materials have been obtained, it advances to the next sub-goal `Craft target: red nether bricks`. This demonstrates how `System 1` decomposes a long-horizon crafting task into sequential milestones aligned with recipe dependencies.

**System 2 (executor).** Figure 15(b) shows `System 2` acting as a low-level executor. Notably, TextCraft provides fine-grained failure feedback (e.g., `Could not find ...`) when prerequisites are missing; `System 2` uses this feedback to adjust subsequent commands (e.g., gathering missing materials before issuing the final `craft` command) while keeping done=`False` until the sub-goal condition is satisfied. Once done=`True`, control returns to `System 1` for the next sub-goal.

## C.2. Hyperparameters

*Table 4.* Hyperparameter configurations for SFT, Offline RL, and Online RL.

| Hyperparameter | SFT | Offline RL | Online RL |
|---|---|---|---|
| *Generation / Decoding* | | | |
| Max new tokens | 32 | 32 | 16 |
| Temperature | 0.7 | 0.7 | 0.7 |
| *Optimization* | | | |
| Optimizer | AdamW | AdamW | AdamW |
| Batch size | 32 | 32 | 32 |
| Micro-batch / GPU | – | 4 | 4 |
| Discount factor $\gamma$ | 0.99 | 0.99 | 0.99 |
| Training epochs | 10 | 20 | – |
| Training steps | – | – | 3500 |
| System 1 LR | $5 \times 10^{-5}$ | – | – |
| System 2 LR (actor) | $1 \times 10^{-4}$ | $1 \times 10^{-4}$ | $1 \times 10^{-6}$ |
| Critic LR | – | $1 \times 10^{-5}$ | $1 \times 10^{-6}$ |
| *Value / Target update* | | | |
| Polyak update ratio $\tau$ | – | 0.2 | 0.2 |
| Target update freq. | – | 2 | 2 |
| Expectile $\tau$ | – | 0.7 | 0.7 |
| *Offline-to-online objective* | | | |
| Strength of imitation $\beta$ | – | 10 | 7 |
| Regularization coefficient $\lambda$ / $\alpha$ | – | 7 | 7 |
| KL regularization coefficient $\eta$ | – | – | 0.02 |
| Clipping | – | 20 | 20 |
| *LoRA* | | | |
| LoRA rank $r$ | 16 | 16 | 16 |
| LoRA $\alpha$ | 32 | 32 | 32 |
| LoRA dropout | 0.05 | 0.05 | 0.05 |

# D. Splits and Evaluation Protocol

## D.1. In-Distribution / Out-of-Distribution Splits

To evaluate generalization under long-horizon interaction, we report results on *ID* and *OOD* splits across benchmarks. Here, the distinction is defined at the task-structure level: an evaluation instance is considered *ID* split if it follows a goal template and interaction pattern seen during training (e.g., the same type of sub-goal decomposition and action schema used to build our role-specific datasets), and it is considered *OOD* split if it follows a held-out template.

**Split Construction.**  For each benchmark, we construct the *ID* and *OOD* splits by matching each evaluation task to the task specifications present in the training corpus. The *ID* split consists of tasks whose goal descriptions (or templates) are observed during training, whereas the *OOD* split consists of held-out tasks that are never used for training in any form. Importantly, even within the *ID* split, evaluation episodes are not fixed: the agent may start from different locations, and object instances may appear in different positions or with different identifiers due to environment stochasticity. As a result, success on the *ID* split still requires robust state grounding and long-horizon control, rather than reproducing memorized action sequences.

**ScienceWorld Split Refinement.**  ScienceWorld provides official split annotations for evaluation; however, for a controlled analysis consistent with the training-set exposure definition above, we additionally construct a strict *ID* and *OOD* partition by checking whether each task specification appears in our training corpus. This refinement is necessary because tasks with different split labels can share overlapping templates or instances, which can blur the boundary between *ID* and *OOD* when evaluation is performed solely using the provided annotations. We therefore use this refined partition for all ScienceWorld results reported in the paper to obtain a clearer assessment of generalization.

## D.2. Prompting Protocol and In-Context Learning

We distinguish between prompt-based baselines that rely on in-context learning (few-shot demonstrations) and fine-tuning-based agents trained by updating model parameters. Unless otherwise stated, we evaluate prompt-based methods with a small set of in-context demonstrations following their original protocols, which increases input length but helps induce the intended interaction format. For fine-tuned agents, including Multi², we do not provide in-context demonstrations at inference time; in particular, Multi² is evaluated in a zero-shot setting and relies on role-specialized training to directly produce sub-goals and executable actions.

**D.3. Task Difficulty**

We define task difficulty in ScienceWorld based on the average interaction length required by our agent to solve each task. Intuitively, tasks that require more environment steps tend to involve long-horizon decision-making and provide more probabilities for compounding errors. Concretely, for each task, we run our agent across multiple evaluation episodes and compute the average number of environment steps taken until termination (success or failure). We then sort tasks by this average interaction length and partition them into three groups of similar size: easy (short), medium, and hard (long).

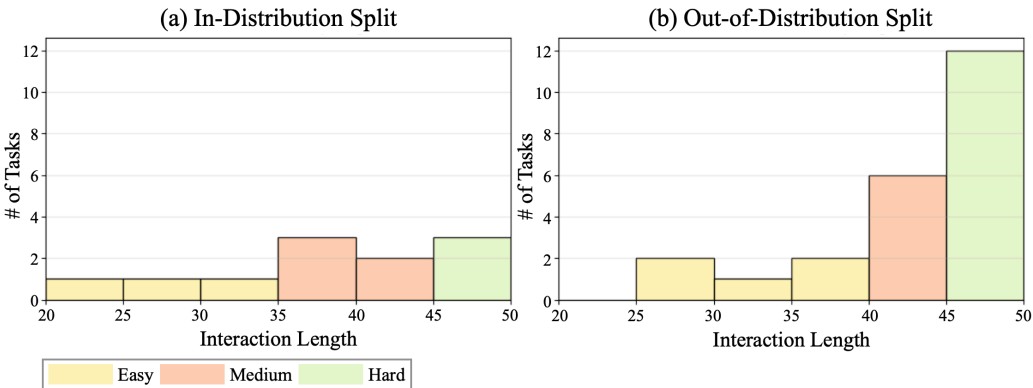

*Figure 16.* Task difficulty based on average interaction length in the ScienceWorld benchmark. For each task, we compute the average number of environment steps taken by our agent across evaluation episodes, and group tasks into Easy/Medium/Hard by quantiles of this length (short/medium/long horizon).

Figure 16 visualizes the distribution of average interaction lengths used to define task difficulty on (a) **In-distribution Split** and (b) **Out-of-distribution Split**. In both splits, easy tasks concentrate in shorter-length bins, medium tasks occupy intermediate ranges, and hard tasks are dominated by the longest interactions. Notably, the *OOD* split is shifted toward longer horizons overall, with a larger mass in the highest-length region (around 45–50 steps) compared to the *ID* split.

# E. Baselines

All baselines use the same backbone LLMs as Multi$^2$ and interact with the identical environment interface (observation formatting, action space, termination conditions, and maximum interaction budget). Across all baselines, we keep (1) the backbone model, (2) the environment wrapper, and (3) the decoding/interaction constraints as consistent as possible.

### E.1. Prompt-Based Baselines

Prompt-based methods do not update model parameters at inference time. To ensure fair comparison, we standardize the environment wrapper and constrain each method to output actions in the same executable format required by the benchmark. Unless otherwise specified, we follow each method's original prompting protocol and include a small number of in-context demonstrations (few-shot examples) to induce the intended reasoning and interaction format.

- ReAct [14]: We follow the canonical Thought–Action–Observation loop, where the agent alternates between intermediate reasoning and an executable action at each step. The agent is prompted to produce an action string compatible with the environment API.

- Reflexion [34]: Reflexion is inherently multi-trial: after each attempt, the agent generates a self-reflection based on episode feedback and stores it in memory to guide subsequent attempts. Accordingly, we report Reflexion under a pass@k protocol that matches its intended evaluation procedure, and we distinguish these results from pass@1 numbers.

- ADaPT [16]: We implement ADaPT as a prompt-based hierarchical agent with a planner and an executor. The planner proposes a sub-goal (or plan sketch) given the task and current observation, and the executor produces step-level actions conditioned on the planner's output.

### E.2. Fine-Tuning-Based Baselines

Fine-tuning baselines update model parameters (via LoRA) using RL-style objectives while keeping the environment interface identical to other methods.

- GRPO [83]: We use GRPO as a single-agent RL fine-tuning baseline. Concretely, a single policy is fine-tuned to maximize environment rewards without introducing an explicit hierarchy; the agent directly maps observations to actions under the same interaction budget used by other methods.

- Glider [15]: Glider is a strong state-of-the-art hierarchical RL framework with a high-level planner and a low-level controller. However, unlike Multi$^2$, it differentiates the two roles primarily through hierarchy prompts while sharing a single parameter-efficient adaptation module across both components, resulting in limited parameter-level decoupling between planning and execution. In addition, Glider trains both levels under a shared training paradigm, whereas Multi$^2$ explicitly treats planning and execution as different optimization problems, using SFT for the planner and offline-to-online RL for the executor. Furthermore, Glider does not continue adapting the low-level executor online, which limits its ability to correct execution-side failures under environment feedback. These differences make Glider a strong and highly relevant comparison point for evaluating the benefits of role-specialized training and executor adaptation in Multi$^2$.

# F. Notation

*Table 5.* Notation of Reinforcement Learning

| Notation | Description | Notation | Description |
|---|---|---|---|
| $\mathcal{S}$ | state space | $\mathbf{s}$ | state |
| $\mathcal{A}$ | action space | $\mathbf{a}$ | action |
| $\mathcal{O}$ | observation space | $\mathbf{o}$ | observation |
| $\mathcal{T}$ | state transition probability | $\Omega$ | observation probability |
| $\mathcal{R}$ | reward function | $r$ | reward |
| $\gamma$ | temporal discounted factor | $\beta$ | scaling factor |
| $\pi_\phi$ | System 1 policy | $\hat{\pi}_\theta$ | SFT-trained System 2 policy |
| $\pi_\theta^{off}$ | System 2 offline-trained policy | $\pi_\theta^{on}$ | System 2 online policy |
| $Q_\omega$ | System 2 Q network | $V_\psi$ | System 2 value network |
| $A$ | advantage function | $\mathcal{J}$ | objective function |
| $\mathcal{L}_\phi$ | System 1 loss function | $\mathcal{L}_\omega$ | Q loss function |
| $\mathcal{L}_\psi$ | value loss function | $\mathcal{L}_\theta$ | policy loss function |
| $L^\tau$ | expectile loss | $\beta$ | strength of imitation |
| $\lambda\ /\ \alpha$ | regularization coefficient | $\eta$ | KL regularization coefficient |

*Table 6.* Notation of Experiment Setups

| Notation | Meaning | Notation | Meaning |
|---|---|---|---|
| $t$ | timesteps | $K$ | textual task description |
| $N$ | the number of tasks | $n$ | $n$-th task |
| $g$ | sub-goal | $H$ | the number of sub-goals |
| $h$ | $h$-th sub-goal | $i$ | $i$-th System 2 data |
| $\mathcal{D}_{sys1}$ | System 1 dataset | $\mathcal{D}_{sys2}$ | System 2 dataset |
| $|\mathcal{D}_{sys1}|$ | the number of System 1 dataset | $|\mathcal{D}_{sys2}|$ | the number of System 2 dataset |
| $M$ | the number of low-level transitions | $\xi$ | the ordered sequence of low-level transition |

# G. Additional Results and Analyses

## G.1. Token Efficiency Analysis

We analyze inference-time token efficiency on the ScienceWorld benchmark. For each method, we report performance (Perform.), average token usage, and token efficiency defined as performance/tokens. To facilitate comparison across methods with different prompting and action formats, we further report normalized token efficiency (TE) by setting ReAct to 1.0 for each split (*ID* and *OOD* splits).

*Table 7.* Token efficiency analysis across *ID* and *OOD* splits with Llama-3.1 8B on ScienceWorld benchmark. Token efficiency is computed as performance/tokens and reported as *normalized* token efficiency where ReAct is set to 1.0.

| Training Type | Structure Type | | Methods | In-Distribution Split | | | Out-of-Distribution Split | | |
| --- | --- | --- | --- | --- | --- | --- | --- | --- | --- |
| | Single | Hier. | | Perform. | Tokens | TE | Perform. | Tokens | TE |
| Prompt-Based | ✓ | | ReAct [14] | 23.23% | 18971.23 | 1.000 | 10.30% | 68031.35 | 1.000 |
| | | ✓ | Reflexion [34] | 22.18% | 18063.16 | 1.003 | 6.97% | 64755.37 | 0.711 |
| | | ✓ | ADaPT [16] | 26.20% | 32449.12 | 0.659 | 12.27% | 63288.85 | 1.281 |
| Fine-Tuning-Based | ✓ | | GRPO [83] | 33.20% | 11145.44 | 2.433 | 4.61% | 9878.00 | 3.083 |
| | | ✓ | Glider [15] | 60.48% | 14716.87 | 3.356 | 34.36% | 20939.88 | 10.838 |
| | | ✓ | Multi² (Proposed) | 67.61% | 12315.42 | 4.483 | 30.68% | 14716.87 | 13.769 |

Table 7 shows that Multi² is consistently the most token-efficient method across both splits and also achieves the highest performance in the *ID* split. On the *OOD* split, the gap widens: Multi² reaches higher performance with substantially lower token usage than prompt-based methods, yielding a larger efficiency gain than Glider and GRPO.

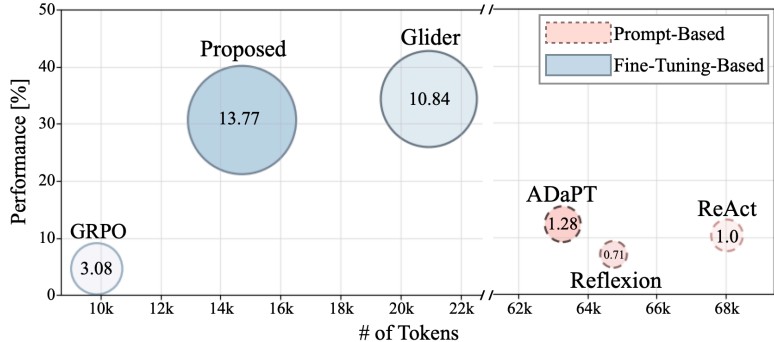

*Figure 17.* Token efficiency at inference time on ScienceWorld benchmark *OOD* split with the Llama-3.1 8B backbone. The $x$-axis denotes token usage, and the $y$-axis denotes performance. Bubble size is proportional to token efficiency, computed as performance/tokens, and the number inside each bubble denotes normalized token efficiency with ReAct set to 1.0. Solid outlines denote fine-tuning-based methods, while dashed outlines denote prompt-based methods.

Figure 17 visualizes token efficiency on the *OOD* split. Multi² attains strong performance with markedly fewer tokens, translating into the largest normalized efficiency gain among all methods. Prompt-based approaches, by comparison, often incur high inference costs due to long in-context demonstrations and multi-turn deliberation, which inflates token usage without proportional performance gains, particularly under distribution shift. In contrast, Multi² benefits from hierarchical, on-demand invocation that avoids unnecessary interaction steps and from RL-based fine-tuning that produces well-formed actions with shorter prompts, improving both performance and token efficiency.

## G.2. Ablation Studies

*Table 8.* Component-wise ablation on ScienceWorld. Each variant removes or modifies one component of the proposed framework.

| Variant | Role-Specific Adapters | Role-Specialized Training | Hierarchical Structure | Proposed Offline Loss | ScienceWorld ID [%] | OOD [%] |
|---|---|---|---|---|---|---|
| Shared Adapter | ✗ | ✓ | ✓ | ✓ | 45.57 | 22.93 |
| RL-SFT | ✓ | ✗ | ✓ | ✓ | 52.05 | 18.15 |
| Only SFT | ✓ | ✗ | ✓ | – | 53.72 | 18.35 |
| Only RL | ✓ | ✗ | ✓ | ✓ | 57.32 | 28.93 |
| Single | – | – | ✗ | ✓ | 56.02 | 26.39 |
| Vanilla-IQL | ✓ | ✓ | ✓ | ✗ | 56.38 | 27.25 |
| Proposed | ✓ | ✓ | ✓ | ✓ | 60.68 | 29.04 |

**Component-Wise Analysis.** Table 8 presents a component-wise ablation on ScienceWorld. We observe that removing individual components leads to consistent performance drops compared to the full model. In particular, eliminating role-specific adapters (Shared Adapter) results in a significant degradation, and removing role-specialized training (RL-SFT, Only SFT, Only RL) also leads to notable declines across both *ID* and *OOD* splits.

In contrast, removing the hierarchical structure (Single) or the proposed offline loss (Vanilla-IQL) leads to relatively smaller but still consistent drops. These results suggest that the gains are cumulative rather than attributable to a single factor. Overall, the strongest performance is achieved by the full combination of parameter decoupling, role-specialized training, hierarchical structure, and the proposed objective design.

**Training Configurations.** To examine whether the role-specialized training design generalizes beyond ScienceWorld, we extend the training-assignment ablation to ALFWorld and TextCraft.

*Table 9.* Cross-benchmark results for the training-assignment ablation.

| Method | Training Assignment System 1 | System 2 | ScienceWorld ID [%] | OOD [%] | ALFWorld ID [%] | OOD [%] | TextCraft Success [%] |
|---|---|---|---|---|---|---|---|
| Only SFT | SFT | SFT | 60.51 | 28.17 | 49.60 | 42.40 | 22.50 |
| RL-SFT | RL | SFT | 48.75 | 22.04 | 53.60 | 50.70 | 12.50 |
| Only RL | RL | RL | 57.62 | 30.47 | 56.20 | 52.60 | 18.00 |
| Proposed | SFT | RL | 67.61 | 30.68 | 57.86 | 56.43 | 35.60 |

Table 9 compares alternative assignments of SFT and RL to System 1 and System 2 across all benchmarks. Overall, the proposed training assignment remains the strongest variant across benchmarks. The swapped RL-SFT setting performs particularly poorly on ScienceWorld and TextCraft, while the single-paradigm variants (*Only SFT* and *Only RL*) remain weaker than the proposed role-specialized design. These results support our claim that assigning SFT to System 1 and RL to System 2 is important for robust long-horizon interaction.

**Adapter Type.** To isolate the contribution of parameter decoupling, we additionally compare a shared adapter with the proposed role-specific adapters while keeping the rest of the training pipeline unchanged.

*Table 10.* Effect of adapter design across additional benchmarks. We compare a single shared adapter against the proposed role-specific adapters while keeping the rest of the training pipeline unchanged.

| Adapter | ScienceWorld | | ALFWorld | | TextCraft |
|---|---|---|---|---|---|
| | ID [%] | OOD [%] | ID [%] | OOD [%] | Success [%] |
| Shared | 52.53 | 24.61 | 48.60 | 48.60 | 11.10 |
| Role-Specific (Proposed) | 67.61 | 30.68 | 57.86 | 56.43 | 35.60 |

Table 10 reports the results on ScienceWorld, ALFWorld, and TextCraft. Across all benchmarks, the proposed role-specific adapters consistently outperform the shared adapter variant. The gain is especially pronounced on TextCraft, and substantial improvements are also observed on ALFWorld and ScienceWorld. These results support our claim that parameter decoupling is important for effective role specialization, beyond the effect of the training objective alone.

**Model Scaling Analysis.** We analyze how model scale affects performance in the ALFWorld interactive environment.

*Table 11.* Success rate on ALFWorld for *ID* and *OOD* splits across Qwen-2.5 model scales. Highlighted cells indicate the best results within each split among the reported scales.

| Model scale | ALFWorld | |
|---|---|---|
| | ID [%] | OOD [%] |
| Qwen-2.5 1.5B | 53.57 | 48.57 |
| Qwen-2.5 3B | 61.43 | 49.29 |
| Qwen-2.5 7B | 62.10 | 64.29 |

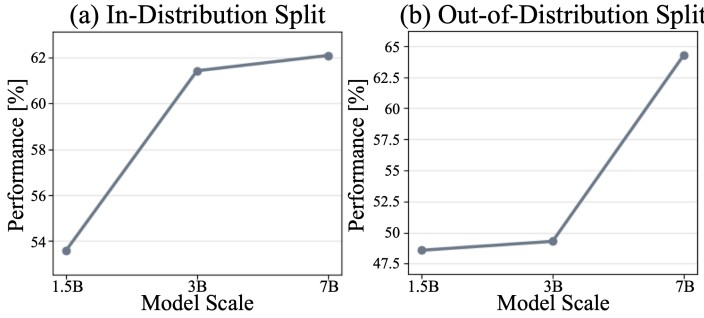

*Figure 18.* Model scaling trends on ALFWorld benchmark across Qwen-2.5 model scales. (a) **In-Distribution Split** and (b) **Out-of-Distribution Split**.

Table 11 reports Multi² results for Qwen-2.5 at three scales (1.5B, 3B, 7B) on ALFWorld under the *ID/OOD* splits. We observe a clear scaling trend, with success improving as model size increases on both splits. Figure 18 summarizes these trends. Notably, the gains are more pronounced under distribution shift: OOD success increases from 48.57 to 64.29 when scaling from 1.5B to 7B.

### G.3. Online Adaptation Analyses

**Performance Analysis.** We analyze how online adaptation affects performance across task difficulty on the ScienceWorld benchmark.

*Table 12.* Online adaptation on ScienceWorld with the Qwen-2.5 3B backbone. We report performance of the offline-trained policy (Before), performance after online adaptation (After), and the relative improvement rate ($\Delta_{rel}$).

| Task Difficulty | In-Distribution Split | | | Out-of-Distribution Split | | |
|---|---|---|---|---|---|---|
| | Before [%] | After [%] | $\Delta_{rel}$ [%] | Before [%] | After [%] | $\Delta_{rel}$ [%] |
| Easy | 76.09 | 81.04 | +6.51 | 36.96 | 42.19 | +14.15 |
| Medium | 63.12 | 68.21 | +8.06 | 27.44 | 27.46 | +0.07 |
| Hard | 58.41 | 59.21 | +1.37 | 16.19 | 17.11 | +5.68 |

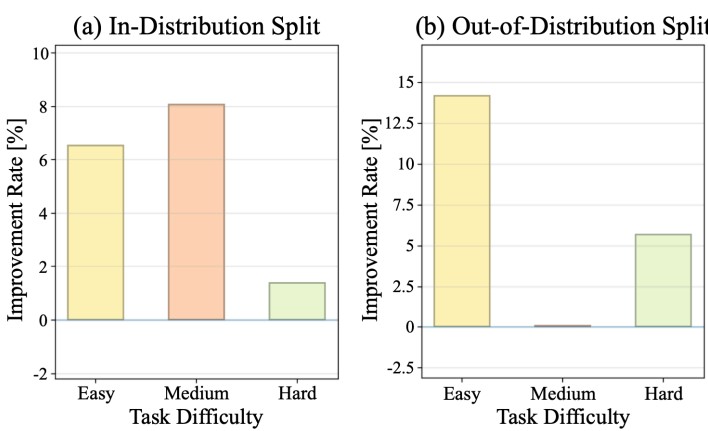

*Figure 19.* Relative improvement rate from online adaptation on ScienceWorld, stratified by task difficulty for **(a) In-Distribution Split** and **(b) Out-of-Distribution Split**, measured relative to the offline-only baseline.

Table 12 reports before/after performance and the relative improvement rate ($\Delta_{rel}$) across task difficulty for both *ID* and *OOD* splits, where $\Delta_{rel}$ is computed as the relative gain over the offline-only baseline. Figure 19 visualizes the same trend and suggests that online adaptation provides a reliable mechanism for self-improvement. Across both splits, online updates consistently improve performance, with the largest gains on easier tasks and generally larger improvements under distribution shift. The agent learns to reduce repetitive action loops, correct local mis-targeting (e.g., selecting the wrong object/container), and improve follow-through by better aligning actions with the current observation.

**Training Setting Analysis.** We provide additional analysis of the online RL stage in terms of training cost, stability, and learning dynamics.

Relative to offline training, the online RL stage introduces only a modest computational overhead, adding approximately 20% additional training time. Despite this, it consistently improves performance from the offline-only setting (60.68 / 29.04) to the online-adapted setting (69.30 / 30.12) on the ScienceWorld *ID / OOD* splits. This indicates a favorable cost–performance trade-off, where a small increase in training time yields a substantial performance gain.

To further understand the behavior of online adaptation, we analyze the training curve on ScienceWorld.

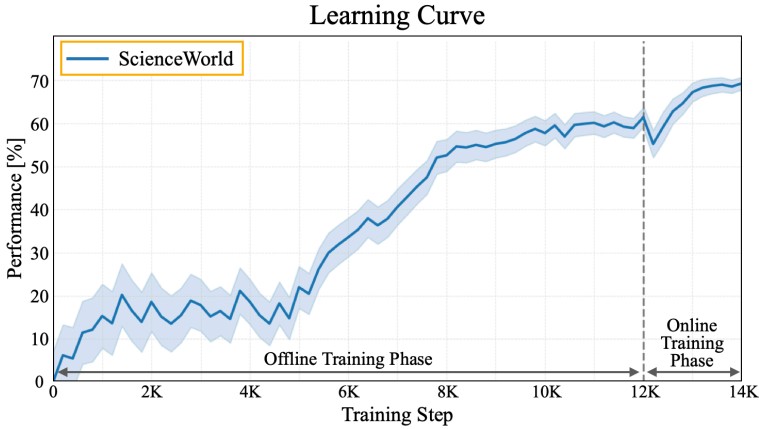

*Figure 20.* Training curve of the offline-to-online RL stage on ScienceWorld.

In Figure 20, we observe a steady improvement in performance during online interaction, without noticeable instability or collapse. This result suggests that online RL yields progressive refinement rather than only an early transient gain, while remaining practically manageable in our setting.

### G.4. Impact of Planner-Generated Sub-Goals

We analyze the impact of planner-generated sub-goals on overall performance by separating planner-related failures from other failed episodes on ALFWorld. Since `System 2` is conditioned on sub-goals produced by `System 1`, errors in sub-goal generation could potentially propagate and degrade downstream execution.

To quantify this effect, we categorize failed episodes based on whether the failure can be attributed to incorrect or suboptimal sub-goals.

*Table 13.* Planner-related failure analysis on ALFWorld.

| Split | Planner-Related Failures | Total Failed Episodes |
|-------|--------------------------|-----------------------|
| ID    | 2 (3.70%)                | 54                    |
| OOD   | 4 (5.88%)                | 68                    |

Table 13 summarizes the results. Planner-related failures are relatively rare, accounting for only $2/54$ (3.70%) of failed episodes on the *ID* split and $4/68$ (5.88%) on the *OOD* split. This indicates that sub-goal errors are not the dominant source of failure in our framework. Instead, most failures arise from downstream execution errors, suggesting that improving the low-level policy is more critical than refining sub-goal generation in the current setting. This observation further supports our design choice of focusing online adaptation on `System 2` while keeping `System 1` fixed.

## G.5. Detailed Case Studies

We provide qualitative rollouts on the ScienceWorld `Find-a-plant` task under the same evaluation setting to illustrate representative failure modes of baselines. Red boxes highlight critical failures, including invalid/rejected commands (e.g., `No known action matches that input`) and non-progressing loops (repeating actions without state change).

**ReAct**

ReAct eventually issues invalid navigation commands (e.g., `open door to living room/foundry`), repeatedly triggering `No known action matches that input` and stalling recovery.

**Task**: Find a(n) plant. First, focus on the thing. Then, move it to the yellow box in the living room

**Observation**: This room is called the foundry. In it, you see:   the agent  a substance called air  a blast furnace, which is turned off. The blast furnace door is closed.   a sink, which is turned off. In the sink is: nothing.   a table. On the table is: nothing. You also see:  A door to the outside (that is open)

**Think**: To solve the task, I need to Move to a new location

**Action**: Open door to outside

**Observation**: The door is already open

**Action**: Go to outside

**Observation**: You move to the outside

**Think**: Now I Move to a new location and Move to the location of the answer box. Next, I need to focus on a non-living thing

**Action**: Focus on cupboard

**Observation**: You focus on the cupboard

**Think**: Now I focus on a non-living thing. Next, I need to move living thing to answer box

**Action**: Move cupboard to sink

**Observation**: You move the cupboard to the sink

**Think**: Now I move living thing to answer box. Next, I need to

**Action**: Open door to living room

**Observation**: No known action matches that input

**Action**: Open door to foundry

**Observation**: No known action matches that input

*Figure 21.* Test case result of ReAct, evaluated on the ScienceWorld `Find-a-plant` task.

**Reflexion**

Reflexion enters a non-progressing loop by repeatedly executing the same action (e.g., `focus on adult cherry tree`) without advancing toward the remaining task requirements.

---

**Task:** Find a(n) plant. First, focus on the thing. Then, move it to the purple box in the living room.

---

**Observation:** This room is called the bathroom. In it, you see:  the agent  a substance called air  a bathtub, which is turned off. In the bathtub is: nothing.  a drawing  a glass cup (containing nothing)  a sink, which is turned off. In the sink is: nothing.  a toilet. In the toilet is: A drain, which is open, a substance called water. You also see:  A door to the kitchen (that is open)

---

**Think:** To solve the task, I need to Move to a new location and Move to the location of the answer box

---

**Action:** Open door to kitchen

---

**Observation:** The door is already open

---

**Action:** Go to kitchen

---

**Observation:** You move to the kitchen

---

**Think:** Now I Move to a new location. Next, I need to focus on a plant

---

**Action:** Focus on adult cherry tree

---

**Observation:** You focus on the adult cherry tree

---

**Action:** Focus on adult cherry tree

---

**Observation:** You focus on the adult cherry tree

---

*Figure 22.* Test case result of Reflexion, evaluated on the ScienceWorld `Find-a-plant` task.

**ADaPT**

Despite producing a plausible high-level plan, ADaPT repeatedly proposes an invalid move (`go to studio`), causing consecutive rejected actions and preventing progress.

---

**Task:** Find a(n) plant. First, focus on the thing. Then, move it to the yellow box in the living room

---

**Observation:** This room is called the foundry. In it, you see:
    the agent, a substance called air, a blast furnace, which is turned off. The blast furnace door is closed. a sink, which is turned off. In the sink is: nothing. a table. On the table is: nothing. You also see: A door to the outside (that is open)

---

**Planner:** [Focus on the plant], [Move the plant to the red box in the living room]

---

**Action:** Open door to hallway

---

**Observation:** The door is now open

---

**Action:** Go to hallway

---

**Observation:** You move to the hallway

---

**Action:** Go to studio

---

**Observation:** No known action matches that input

---

**Action:** Go to studio

---

**Observation:** No known action matches that input

---

*Figure 23.* Test case result of ADaPT, evaluated on the ScienceWorld `Find-a-plant` task.

**GRPO**

GRPO reaches a plausible target (greenhouse and a plant) but then repeatedly performs the same interaction (`focus on adult avocado tree`), indicating a loop rather than completing the full multi-step objective.

---

**Task**: Find a(n) plant. First, focus on the thing. Then, move it to the purple box in the bathroom

---

**Observation**: This room is called the living room. In it, you see:
    the agent, a substance called air, a book shelf (containing nothing). a chair. On the chair is: nothing. a couch. On the couch is: a white pillow. a desk. On the desk is: a drawer. a painting. You also see: A door to the hallway (that is closed)

---

**Action**: Open door to hallway

---

**Observation**: The door is now open

---

**Action**: Go to hallway

---

**Observation**: You move to the hallway

---

**Action**: Open door to greenhouse

---

**Observation**: The door is now open

---

**Action**: Go to greenhouse

---

**Observation**: You move to the greenhouse. In it, you see: the agent. a substance called air, a bee hive. The bee hive door is closed. a flower pot 1 (containing a banana tree in the adult stage, soil), a flower pot 2 (containing a avocado tree in the adult stage, soil), a sink, which is turned off. In the sink is: nothing. You also see: A door to the hallway (that is open)

---

**Action**: Focus on adult avocado tree

---

**Observation**: You focus on the avocado tree

---

**Action**: Focus on adult avocado tree

---

**Observation**: You focus on the avocado tree

---

**Action**: Focus on adult avocado tree

---

**Observation**: You focus on the avocado tree

---

*Figure 24.* Test case result of GRPO, evaluated on the ScienceWorld `Find-a-plant` task.

**Glider**

Glider follows a subtask sequence but issues an invalid manipulation command (e.g., `pick up flower pot 6`), leading to rejected actions and breaking execution of the intended sub-goal.

> **Task**: Find a(n) plant. First, focus on the thing. Then, move it to the orange box in the bathroom

> **Observation**: This room is called the art studio. In it, you see: the agent, a substance called air, a large cupboard. The large cupboard door is closed. a table. On the table is: a bowl (containing nothing). a wood cup (containing red paint), a wood cup (containing yellow paint), a wood cup (containing blue paint) You also see: A door to the hallway (that is closed)

> **Subtask**: Find a plant and focus it

> **Action**: Open door to hallway

> **Observation**: The door is now open

> **Action**: Go to hallway

> **Observation**: You move to the hallway

> **Action**: Open door to greenhouse

> **Observation**: The door is now open

> **Action**: Go to greenhouse

> **Observation**: You move to the greenhouse

> **Action**: look around

> **Observation**: This room is called the greenhouse. In it, you see: the agent, a substance called air, a bee hive. The bee hive door is closed. a flower pot 1 (containing a banana tree in the adult stage, soil), a flower pot 2 (containing nothing), a flower pot 4 (containing a pea plant in the adult stage with a tall height, soil), a flower pot 8 (containing a avocado tree in the adult stage, soil), a flower pot 9 (containing nothing), a jug (containing nothing), a sink, which is turned off. In the sink is: nothing. You also see: A door to the hallway (that is open), A door to the outside (that is closed)

> **Action**: focus on adult banana tree

> **Observation**: You focus on the banana tree.

> **Subtask**: Navigation to bathroom with the plant

> **Action**: pick up flower pot 6

> **Observation**: No known action matches that input.

> **Action**: open door to hallway

*Figure 25.* Test case result of Glider, evaluated on the ScienceWorld `Find-a-plant` task.

