# OpenReview forum: "$\texttt{Multi}^2$: Hierarchical Multi-Agent Decision-Making with LLM-Based Agents in Interactive Environments"
_ICML.cc/2026/Conference — ICML 2026 regular_

### Official Review · Reviewer_hpqy · 2026-02-28

**Soundness:** 3
**Presentation:** 3
**Significance:** 2
**Originality:** 2
**Overall Recommendation:** 3
**Confidence:** 3

**Summary:**

This paper proposes Multi2, a hierarchical multi-agent decision-making framework for long-horizon interactive environments, targeting two key issues in LLM-based agents: goal drift and token inefficiency. The framework explicitly decomposes agent behavior into two complementary roles: System 1 (high-level agent) generates context-aware subgoals via SFT on a high-level dataset, while System 2 (low-level agent) executes atomic actions in interactive environments via an offline-to-online RL pipeline. Both systems share a pretrained backbone but use separate LoRA adapters for role specialization. In the offline phase, System 2's actor loss combines exponential-advantage-weighted behavior cloning with a policy-anchored advantage term; the online phase introduces KL regularization to constrain the online policy near the offline policy, preventing mode collapse and enabling stable self-improvement. Experiments are conducted on three text-based interactive environments (ScienceWorld, ALFWorld, TextCraft) with three backbone models (Qwen-2.5 3B, Mistral 7B, Llama-3.1 8B). Key results: (i) Multi2 achieves the best performance on most backbone-split combinations; (ii) Multi2 is more token-efficient; (iii) Multi2's advantage over baselines grows with task difficulty; (iv) ablations validate the contribution of role-specialized training, loss design, and online adaptation. The paper also releases three hierarchical benchmark datasets.

**Compliance With Llm Reviewing Policy:**

Affirmed.

**Final Justification:**

I thank the authors for the thorough two-round rebuttal. The responses were constructive and addressed several of my original concerns:

- Q-value inflation (Q1/W3): The authors provided a diagnostic tracking the gap between critic values on dataset vs. policy-sampled actions, showing the gap remains stable rather than escalating. While this does not fully rule out overestimation, it is a reasonable empirical response. I accept the authors' framing of this as an intentional conservatism-improvement trade-off.
- Controlled ablation (Q2/W2): The single-adapter vs. role-specific adapter ablation was provided and shows a clear performance gap, confirming that parameter decoupling contributes meaningfully beyond the training objective alone. This resolves my original concern.
- System 1 fine-tuning (Q3/W1): The failure analysis showing that planner-related errors account for a small fraction of failures (3.7% ID, 5.9% OOD) provides reasonable justification for freezing System 1. I accept this as a pragmatic design choice, though the limitation remains.

Despite these clarifications, my main reservation concerns originality and significance. The individual components of Multi2—hierarchical goal decomposition, separate LoRA adapters for role specialization, IQL-based offline RL, KL-regularized online adaptation—are well-established techniques. The paper's contribution lies in their integration for long-horizon text-based agent tasks, but I find this integration to be more of a reasonable engineering combination than a conceptually novel advance. The second-round rebuttal frames novelty around addressing "objective drift," but the mechanism for doing so (hierarchical decomposition + executor RL) follows a fairly standard recipe in hierarchical RL. The reported improvements, while consistent, are also modest on several splits (e.g., OOD ScienceWorld: 30.30 → 30.68).

I maintain my score of Weak Reject. The paper is technically sound, well-executed, and the released datasets are a useful community contribution. However, the limited conceptual novelty beyond combining known components does not, in my assessment, meet the bar for acceptance at this venue.

**Key Questions For Authors:**

1. The policy-anchored advantage term in the offline actor loss (Eq. 6) evaluates Q on actions from the current policy, which contradicts IQL's core principle of avoiding OOD queries. Please analyze whether Q-value inflation occurs during training and discuss whether this term introduces overestimation. If so, have you considered alternative designs (e.g., evaluating advantage only on dataset actions, or using a conservative Q estimate for the anchoring term)?
2. Please provide a controlled ablation: keeping Multi2's training pipeline unchanged, replace the two separate LoRA adapters with a single shared adapter to isolate the contribution of parameter decoupling to performance gains.
3. Have you considered fine-tuning System 1 during the online RL phase? If System 1's subgoal generation is suboptimal in certain environments, does the current framework have any mechanism to detect and mitigate this?

**Limitations:**

yes

**Strengths And Weaknesses:**

### Strengths

- The role-specialized training design is well-motivated and empirically validated. The ablation study (Table 2, Figure 6(a)) clearly shows that swapping training paradigms (RL for System 1, SFT for System 2) yields the worst configuration, pure RL for high-level planning is unstable with high variance, and SFT-only System 2 is brittle under environmental variation. These ablations effectively support the core design rationale that SFT suits semantic planning while RL suits environmental interaction.
- Experimental coverage is broad. Three benchmark environments (ScienceWorld's stateful science experiments, ALFWorld's household manipulation, TextCraft's combinatorial crafting) cover complementary interaction patterns and reward structures; three backbone models of different scales and families verify framework generality.
- The token efficiency analysis (Figure 4, Table 6) provides a practical perspective. Multi2 reduces inference token usage through hierarchical on-demand invocation (System 1 is only called upon subgoal completion) and RL-tuned formatting (well-formed actions without long prompts), achieving better performance with significantly fewer tokens. This has direct implications for practical deployment.
- The paper releases processed hierarchical datasets, extraction rules, and training code. The dataset construction uses a deterministic code-based pipeline rather than LLM-dependent extraction, improving reproducibility.

### Weaknesses

- System 1 is trained only via SFT and relies entirely on the quality of subgoals in offline expert data, receiving no feedback from environment interaction throughout the entire training process. If the expert data contains suboptimal subgoal decompositions (or ones that do not adapt to specific environment instances), System 1 has no mechanism for self-correction. The paper freezes System 1 during the online phase and updates only System 2, but does not discuss this design trade-off or explore the possibility of jointly fine-tuning System 1 online.
- A key difference between Multi2 and Glider is the use of separate LoRA adapters (Multi2) vs. a single shared adapter (Glider), but a controlled ablation isolating this factor is missing: keeping Multi2's training pipeline unchanged while replacing the two separate adapters with a single shared one. Without this, the current experiments cannot distinguish whether performance gains come from the improved training algorithm or from parameter decoupling.
- The offline RL loss design contains an internal contradiction regarding overestimation. The Q/V learning (Eqs. 4-5) follows IQL's design: Q's TD target uses V (rather than max_a Q) to avoid querying OOD actions during bootstrapping, and V uses expectile regression (τ=0.7) for implicit maximization—IQL's core design principle is to never evaluate Q on out-of-distribution actions, thereby fundamentally avoiding overestimation. However, the policy-anchored advantage term λA(ot, π_θ(at|ot;gh); gh) in the offline actor loss (Eq. 6) evaluates A = Q − V on actions from the **current policy**, which may deviate from the offline data distribution, causing Q to be queried on OOD actions. This directly undermines IQL's overestimation protection. The paper does not discuss this contradiction or provide any diagnostic analysis of Q-value estimation accuracy (e.g., whether Q-values inflate during training).

---

> ### Author Rebuttal · Authors · 2026-03-30
>
> We thank the reviewer for the insightful feedback. Below, we address the reviewer's concerns and questions.
>
> ---
>
> ### **Please analyze whether Q-value inflation occurs during training (Q1, W3)**
>
> We agree that the policy-anchored advantage term in Eq. (6) is **less conservative than vanilla IQL**. The critic uses dataset-supported TD targets and expectile value learning, but the actor-side term evaluates **policy-sampled actions**. Therefore, our method does not preserve the strict IQL-style property of never querying OOD actions.
>
> This is an intentional trade-off: strictly in-sample updates are more conservative, but they can also tie policy improvement too closely to the exact support of the offline dataset, thereby **limiting improvement** beyond observed trajectories. More broadly, offline RL often involves a trade-off between conservatism and policy improvement beyond the dataset support [1–3].
>
> To examine whether this causes harmful overestimation, we monitor critic values on **dataset actions** $Q(s,a_D)$ and **policy-sampled actions** $Q(s,\pi(s))$, and track the gap $Q(s,\pi(s)) - Q(s,a_D)$ over training for both vanilla IQL and our method:
>
> |Method|1–3K|4–6K|7–9K|10–12K|
> |:--|:--:|:--:|:--:|:--:|
> |Vanilla-IQL|-0.09|-0.04|0.14|0.18|
> |Proposed|0.22|0.28|0.25|0.23|
>
> We use this as a *diagnostic proxy* for conservatism/overestimation. While this metric does not rule out all forms of overestimation, we do *not* observe an escalating optimism trend in our method: the gap remains relatively stable and slightly decreases in the later phase (0.22 → 0.28 → 0.25 → 0.23), rather than widening progressively over training.
>
> This less conservative update is also empirically beneficial:
>
> |Method|ScienceWorld (ID / OOD) [%]|ALFWorld (ID / OOD) [%]|TextCraft [%]|
> |:--|:--:|:--:|:--:|
> |Vanilla-IQL|58.78 / 30.30|39.29 / 30.70|30.50|
> |**Proposed**|**67.61** / **30.68**|**57.86** / **56.43**|**35.60**|
>
> Overall, these results suggest that the policy-anchored term makes the update less conservative and improves downstream performance, without clear evidence of runaway Q-value inflation in this diagnostic.
>
> [1] S. Fujimoto, et al., "A Minimalist Approach to Offline Reinforcement Learning," in *NeurIPS*, 2021.
>
> [2] Z. Wang, et al., "Diffusion Policies as an Expressive Policy Class for Reinforcement Learning," in *ICLR*, 2023.
>
> [3] Y. Qing, et al., "A2PO: Towards Effective Offline Reinforcement Learning from an Advantage-aware Perspective," in *NeurIPS*, 2024.
>
> ---
>
> ### **Please provide a controlled ablation (Q2, W2)**
>
> Thank you for this helpful suggestion. To directly address this concern, we conduct an additional controlled ablation in which we replace the two **role-specific LoRA adapters** with a **single shared adapter**.
>
> |Adapter|ScienceWorld (ID / OOD) [%]|ALFWorld (ID / OOD) [%]|TextCraft [%]|
> |:--|:--:|:--:|:--:|
> |Single|52.53 / 24.61|48.60 / 48.60|11.10|
> |**Role-Specific (Proposed)**|**67.61** / **30.68**|**57.86** / **56.43**|**35.60**|
>
> Role-specific adapters consistently outperform the single-adapter variant across all benchmarks, indicating that **parameter decoupling is important for effective role specialization**, beyond the training objective alone.
>
> ---
>
> ### **Have you considered fine-tuning System 1 during online RL? (Q3, W1)**
>
> In the current version of Multi², we do *not* fine-tune System 1 during online interaction. This is a deliberate design choice: we keep System 1 as a stable high-level planner and focus online adaptation on System 2, where feedback is more immediate and informative.
>
> Our ALFWorld analysis suggests that **planner-related errors are not the dominant bottleneck**:
>
> - Among failed ALFWorld episodes, only 2 / 54 *ID* failures (3.70%) and 4 / 68 *OOD* failures (5.88%) were attributable to clearly problematic sub-goals.
>
> - Execution-related failures were much more common and were reduced by online RL (ID: 52→46, OOD: 64→52).
>
> Most failures arise from *downstream execution* rather than from clearly problematic sub-goals produced by System 1. In addition, System 1 can be re-invoked using the updated interaction context after sub-goal termination, which can **mitigate imperfect sub-goals**.
>
> We agree that jointly fine-tuning System 1 during online RL is an interesting and important future direction—especially in *OOD* settings where sub-goal decompositions may be suboptimal. In the current version, however, we chose to prioritize **planner stability** and adapt System 2, which appears to be the more significant bottleneck in our present analysis.
>
> ---
>
> **Final remark**
>
> We are grateful for the reviewer’s insightful comments and the opportunity to further clarify our work. We hope that the additional discussion and diagnostics on Q-value inflation, the ablation, and the analysis of System 1 have sufficiently addressed the reviewer’s questions. If the reviewer finds that these additions have addressed the concerns, we would be grateful if this could be reflected in the final assessment.

---

> > ### Author Rebuttal · Reviewer_hpqy · 2026-04-03
> >
> > Thank you to the authors for the rebuttal. The response addressed some of my concerns and clarified several points. However, I still find the overall novelty of the paper limited, and this remains my main reservation. Therefore, I will maintain my Weak Reject score.

---

> > > ### Author Response · Authors · 2026-04-03
> > >
> > > We are glad to hear that our previous response helped address several of your concerns. Thank you again for your thoughtful comments and for engaging so carefully with our paper.
> > >
> > > In our earlier rebuttal, we aimed to directly address the main technical questions raised in the original review.
> > >
> > > - **[Q1, W3] Q-value / RL formulation**: We clarified the design of Eq. 6 and its relation to IQL, including its behavior under *OOD* actions.
> > > - **[Q2, W2] Controlled ablations**: We added explicit controlled studies to isolate the contributions of role-specific adapters and role-specialized training.
> > > - **[Q3, W1] Training strategy**: We clarified the role of SFT versus RL and showed that role-aligned optimization is critical for performance.
> > >
> > > We are grateful that these additional analyses and clarifications were helpful. Since novelty appears to be the main remaining question, we would be happy to briefly summarize how we view the paper’s contribution.
> > >
> > > ---
> > >
> > > ## **Novelty of our paper**
> > > We view the novelty of this work not as a completely new primitive in isolation, but in **how it addresses a concrete unresolved problem** in interactive LLM agents.
> > >
> > > ### **Why this paper is needed: objective drift as a persistent, unresolved failure mode**
> > >
> > > |Method|Performance [%]|Horizon: Short|Medium|Long|
> > > |:-|-|-:|:-:|:-:|
> > > |ReAct||33.46|17.98|12.08|
> > > |Glider||68.03|58.41|49.44|
> > > |**Multi² (Proposed)**||**70.71**|**70.83**|**67.29**|
> > >
> > > As shown in Section 1, the main weakness of prior methods emerges at **longer horizons**: the performance drops substantially as the interaction becomes longer, whereas **Multi² remains stable**. This pattern is also reflected in the qualitative failure cases:
> > >
> > > |Method|Failure Pattern|Structural Limitation|
> > > |:-|:-|:-|
> > > |ReAct|Repeats invalid actions such as `open door to living room` or `foundry`|Flat prompting makes it harder to maintain goal consistency.|
> > > |Glider|Follows subtask structure, but the executor issues invalid object-level actions|Shared role parameterization may limit clean specialization.|
> > > |Multi² (Proposed)|Maintains sub-goal-anchored execution and recovers from invalid states|Explicit sub-goal anchoring and adaptive executor training reduce these execution-side failures.|
> > >
> > > In our setting, **long-horizon objective drift** often appears as **execution-side failures** that accumulate over multi-turn interaction. Accordingly, Multi² is designed explicitly for *robust long-horizon interaction*, rather than for benchmark improvement in the abstract.
> > >
> > > ### **Why execution is the bottleneck**
> > > Our analysis suggests that **long-horizon failures are driven mainly by the execution stage** rather than by the planner. In ALFWorld, planner-related failures are relatively rare (ID: 2/54, OOD: 4/68), which motivated us to prioritize executor adaptation.
> > >
> > > This is also consistent with a limitation noted in Glider: with parameter sharing, the low-level policy cannot be cleanly separated during online adaptation (Appendix A). This provides further rationale for our design choice of **separate adapters and executor-focused online adaptation**.
> > >
> > > ### **Q-guided policy improvement for this bottleneck**
> > > The main methodological contribution is a **role-aligned hierarchical framework** that separates high-level planning from low-level execution and aligns them with different optimization strategies: SFT for System 1 and offline-to-online RL for System 2.
> > >
> > > Beyond hierarchy itself, the offline actor loss (Eq. 6) introduces a **policy-anchored advantage term for Q-guided policy improvement**, while the online loss (Eq. 7) uses KL-regularized refinement to stabilize adaptation. We view this combination as important because long-horizon failures are not resolved by decomposition alone; they also require robust online adaptation.
> > >
> > > ### **Practical contribution in token-efficient long-horizon interaction**
> > > Multi² also improves token efficiency in long-horizon interaction through hierarchical, on-demand invocation and executor-side RL tuning.
> > >
> > > |Method|Token Efficiency|
> > > |:-|:-:|
> > > |ReAct|1.000|
> > > |Reflexion|1.003|
> > > |ADaPT|0.659|
> > > |GRPO|2.433|
> > > |Glider|3.356|
> > > |**Multi² (Proposed)**|**4.483**|
> > >
> > > ### **Community contribution: reproducible datasets across multiple benchmarks**
> > > Beyond the method itself, the paper also contributes hierarchical datasets and training code for ScienceWorld, ALFWorld, and TextCraft.
> > >
> > > A practical challenge in hierarchical agent research is that dataset construction often requires **substantial preprocessing and design choices**, which makes fair comparison difficult. We expect this release to help future researchers directly train hierarchical agents, and more broadly support progress on long-horizon LLM-based agents.
> > >
> > > ---
> > >
> > > **Final remark**
> > >
> > > We hope the above clarification helps address the reviewer's concern regarding novelty. If the reviewer feels that the remaining concern has been clarified, we would greatly appreciate reconsideration of the score in the final assessment.

---

### Official Review · Reviewer_DTRV · 2026-03-11

**Soundness:** 3
**Presentation:** 3
**Significance:** 2
**Originality:** 1
**Overall Recommendation:** 4
**Confidence:** 3

**Summary:**

This paper proposes Multi2, a hierarchical framework for long-horizon interactive LLM agents. The method separates planning and execution into System 1, a high-level planner to generate sub-goals, and System 2, a low-level executor. Both share the same backbone but use separate LoRA adapters. Experiments on ScienceWorld, ALFWorld, and TextCraft report improvements over ReAct, Reflexion, ADaPT, GRPO, and Glider in both overall accuracy and token efficiency.

**Compliance With Llm Reviewing Policy:**

Affirmed.

**Final Justification:**

Thank you for the thorough follow-up. The adapter ablation, cross-dataset ablations, matched token-efficiency comparison, and stability curves address my technical concerns, which is why I raised my score to Weak Accept during the rebuttal and will keep it there. I still disagree on the two conceptual points. I see Multi^2 as a strong engineering advance over Glider rather than a new conceptual framework, and I continue to find the multi-agent framing misleading; this is a clean hierarchical decomposition, not a multi-agent system. Just because two other papers (including one workshop) have gotten away with the framing does not change the fact that it is incorrect. Words mean things.

That said, the practical contributions are clear, especially the efficiency gains and the dataset release, and I appreciate the care you put into the response.

**Key Questions For Authors:**

1. Have your conducted an ablation comparing separate role-specific adapters versus a shared adapter? This is a central claimed distinction from Glider.
2. What are the token-efficiency results when all methods use matched inference protocols?
3. Have you ran the ablations on the other environments? They seem restricted to ScienceWorld in the current version of the manuscript.
4. Does System1 have the ability to replan when System2 encounters failure modes? This seems important in OOD scenarios.
5. Can you release stability curves during the online training? Are the results over multiple seeds or just the single best. Some insight on this process would be appreciated.

**Limitations:**

yes

**Strengths And Weaknesses:**

## Strengths

- Table 1 shows strong improvements over Glider & other baselines on most settings, especially ALFWorld and TextCraft.
- The planner/executor decomposition is intuitive & the methodology is broadly written clearly.
- Releasing the datasets and code is a useful contribution to the community, and the use of an anonymous project page rather then promising to do it after reviews is appreciated.

## Weaknesses

- The paper generally reads as proposing reasonable engineering improvements over Glider: train the planner with SFT, train the executor with RL, and adding online adaptation. However, conceptually, the ideas are not novel.
- The "multi-agent" framing is misleading. The two modules share the same backbone and differ only through role-specific adapters/ prompts. Hierarchical yes, multiagent, no.
- The paper claims an advantage over Glider based on using separate adapters rather than a shared adapter with prompt-based role separation, but there is no ablation actually comparing shared versus separate adapters. For a central claim, this omission is significant.
- Most ablations are restricted to ScienceWorld. For a paper making broad claims about long-horizon reliability, ablations should span all datasets tested.

---

> ### Author Rebuttal · Authors · 2026-03-30
>
> We thank the reviewer for the detailed and constructive feedback. Below, we address each question and concern.
>
> ---
>
> ### **Differentiation of the Proposed Framework (W1)**
>
> Our contribution is a **role-specialized hierarchical training-and-adaptation framework** for long-horizon interactive environments. We summarize the main differences below:
>
> |Aspect|Glider|Multi²|
> |:---|:---:|:---:|
> |Hierarchical Structure|✔|✔|
> |Role-Specific Adapters|✘|✔|
> |Planner Training|SFT + Offline RL|**SFT**|
> |Executor Training|SFT + Offline RL|**Offline-to-Online RL**|
> |Self-Improvement of Executor|✘|✔|
> |Proposed Objective Design|✘|✔|
>
> Unlike Glider, we do not view hierarchy alone as sufficient. A key technical limitation of Glider is that the planner and executor are still coupled through a **shared prompt-conditioned policy**. In addition, Glider adapts only the high-level policy online, while freezing the low-level skills, which limits its ability to correct **execution-side failures** under environment feedback. Multi² is motivated precisely by these limitations: we treat planning and execution as **different optimization problems** and continue adapting the executor online.
>
> ---
>
> ### **Have you compared separate role-specific adapters with a shared adapter? (Q1, W3)**
>
> We conduct an ablation replacing the two role-specific adapters with a single shared adapter.
>
> |Adapter|ScienceWorld (ID / OOD) [%]|ALFWorld (ID / OOD) [%]|TextCraft [%]|
> |:---|:---:|:---:|:---:|
> |Single|52.53 / 24.61|48.60 / 48.60|11.10|
> |**Role-Specific (Proposed)**|**67.61** / **30.68**|**57.86** / **56.43**|**35.60**|
>
> Role-specific adapters consistently outperform the single-adapter variant across all benchmarks, supporting the importance of **parameter decoupling**.
>
> ---
>
> ### **What are the token-efficiency results when all methods use matched inference protocols? (Q2)**
>
> We agree that token-efficiency should be compared under matched inference protocols. Because prompt-based methods inherently incur large in-context prompt overhead, we report **matched-protocol token-efficiency (TE) within the fine-tuning-based group**, where all methods are evaluated **zero-shot without in-context prompts**.
>
> |Method|ID [%]|ID Tokens|ID TE|OOD [%]|OOD Tokens|OOD TE|
> |:---|:---:|:---:|:---:|:---:|:---:|:---:|
> |GRPO|33.20|11145.44|2.43|4.61|9878.00|3.08|
> |Glider|60.48|14716.87|3.36|34.36|20939.88|10.84|
> |**Multi² (Proposed)**|**67.61**|12315.42|**4.48**|30.68|14716.87|**13.77**|
>
> Multi² achieves the **highest token efficiency among fine-tuning-based methods** on both *ID* and *OOD*.
>
> ---
>
> ### **Have you run the ablations on the other environments? (Q3, W4)**
>
> Thank you for this insightful comment. We extend the ablations of **training-assignment** and **offline-loss** to all benchmarks.
>
> |Method|ScienceWorld (ID / OOD) [%]|ALFWorld (ID / OOD) [%]|TextCraft [%]|
> |:---|:---:|:---:|:---:|
> |Vanilla-IQL|58.78 / 30.30|39.29 / 30.70|30.50|
> |Only SFT|60.51 / 28.17|49.60 / 42.40|22.50|
> |RL-SFT|48.75 / 22.04|53.60 / 50.70| 12.50|
> |Only RL|57.62 / 30.47|56.20 / 52.60|18.00|
> |**Proposed**|**67.61** / **30.68**|**57.86** / **56.43**|**35.60**|
>
> The **proposed full design remains the strongest overall variant**, and removing the proposed training design or loss generally degrades performance.
>
> ---
>
> ### **Does System 1 have the ability to replan? (Q4)**
>
> **System 1 can be re-invoked when the current sub-goal is terminated without completion**, using the updated interaction context to generate the next sub-goal. In this sense, the framework supports **implicit replanning**.
>
> To quantify this behavior, we analyze ALFWorld failure cases. Replanning occurs in 16/54 (29.63%) failed tasks on *ID* and 31/68 (45.59%) on *OOD*, indicating that the framework supports implicit replanning and that such high-level re-invocation is used more often in OOD settings.
>
> ---
>
> ### **Can you release stability curves during the online training? (Q5)**
>
> We thank the reviewer for this helpful suggestion. To address this, we additionally report **mean ± one standard deviation over five random seeds** below.
>
> - **Multiple Seeds Result**
>
> |ScienceWorld ID [%]|OOD [%]|
> |:---:|:---:|
> |69.30 ± 1.98|30.12 ± 0.96|
>
> These results suggest that the online-stage gains are **reasonably stable across runs**.
>
> - **Training-Curve Summary**
>
> |Timesteps|Performance [%]|
> |:---:|:---:|
> |0|61|
> |400|56|
> |800|59|
> |1200|64|
> |1500|68|
> |1800|69|
>
> The overall trend is upward, suggesting that online RL provides **progressive refinement rather than an unstable transient effect**. Seed-wise curves are available on the project page https://anonymous-projectpage.github.io/Multi-Square/.
>
> ---
>
> **Final remark**
>
> We deeply appreciate the reviewer’s careful reading and constructive feedback. If the reviewer finds that these clarifications and added results have addressed the concerns, we would be thankful if this could be taken into consideration in the final evaluation.

---

> > ### Author Rebuttal · Reviewer_DTRV · 2026-03-31
> >
> > I thank the authors for their rebuttal. My technical concerns have been satisfied. I still disagree with the multi-agent framing and am not convinced of the novelty, so I will limit my increase in score to a 4.

---

> > > ### Author Response · Authors · 2026-04-01
> > >
> > > Thank you for your follow-up. We understand that the remaining questions concern **(1) the novelty of the work** and **(2) the appropriateness of the multi-agent framing**. We address each point below.
> > >
> > > ---
> > >
> > > ## **On the novelty of our work**
> > > Multi² introduces an empirically grounded hierarchical training framework targeting a concrete failure mode: **long-horizon objective drift**.
> > >
> > > ### **Why this paper is needed: objective drift as a persistent, unresolved failure mode**
> > > |Method|Performance [%]|Horizon: Short|Medium|Long|
> > > |:-|-|-:|:-:|:-:|
> > > |ReAct||33.46|17.98|12.08|
> > > |Glider||68.03|58.41|49.44|
> > > |**Multi² (Proposed)**||**70.71**|**70.83**|**67.29**|
> > >
> > > As discussed in Section 1, the main gap is at longer horizons, where prior methods degrade substantially but Multi² remains stable. This is also consistent with representative failure patterns:
> > >
> > > |Method|Failure Pattern|Structural Limitation|
> > > |:-|:-|:-|
> > > |ReAct|Repeats invalid actions such as `open door to living room` or `foundry`|Flat prompting makes it harder to maintain goal consistency.|
> > > |Glider|Follows subtask structure, but the executor issues invalid object-level actions|Shared role parameterization may limit clean specialization.|
> > > |Multi² (Proposed)|Maintains sub-goal-anchored execution and recovers from invalid states|Explicit sub-goal anchoring and adaptive executor training reduce these execution-side failures.|
> > >
> > > In our setting, long-horizon objective drift often manifests through execution-side failures. Our goal is to address this *systematic failure mode* rather than simply outperform baselines.
> > >
> > > ### **Why execution is the bottleneck**
> > > Our analysis suggests that **long-horizon failures are dominated by execution**. In ALFWorld, planner-related failures are rare (ID: 2/54, OOD: 4/68), which motivates prioritizing executor adaptation.
> > >
> > > This is also consistent with a limitation noted by Glider itself: because of parameter sharing, the low-level policy cannot be strictly fixed during online adaptation (Appendix A). This further supports our use of **separate adapters and executor-focused online adaptation**.
> > >
> > > ### **Novel actor loss design**
> > > The gains of Multi² are not explained by hierarchical decomposition alone. Beyond this, we also contribute *objective-level design choices*. The offline actor loss (Eq. 6) introduces a **policy-anchored advantage term**, which encourages policy improvement beyond pure dataset imitation while remaining grounded by the critic. Our ablation study shows that this design yields clear gains in long-horizon task performance. The proposed online loss (Eq. 7) also plays an important role in preventing mode collapse during online adaptation.
> > >
> > > ### **Token efficiency as a practical contribution**
> > > Multi² achieves the highest token efficiency.
> > > |Method|Token Efficiency|
> > > |:-|:-:|
> > > |ReAct|1.000|
> > > |Reflexion|1.003|
> > > |ADaPT|0.659|
> > > |GRPO|2.433|
> > > |Glider|3.356|
> > > |**Multi² (Proposed)**|**4.483**|
> > >
> > > ### **Community contribution: reproducible hierarchical datasets**
> > > We also contribute **new hierarchical datasets for long-horizon interactive agent tasks**. To the best of our knowledge, this is the *first release* that provides hierarchical training data across multiple benchmarks: ScienceWorld, ALFWorld, and TextCraft. These datasets were not previously available, and their absence made this line of research substantially harder to pursue.
> > >
> > > A practical challenge in hierarchical agent research is that dataset construction often requires *substantial preprocessing and design choices*, which makes fair comparison difficult. We expect this release to help future researchers directly train hierarchical agents, and more broadly support progress on long-horizon LLM-based agents.
> > >
> > > ---
> > >
> > > ## Appropriateness of the multi-agent framing
> > > We appreciate this point and agree that the term "multi-agent" can be read more strongly than what our architecture strictly instantiates.
> > >
> > > Our use of the term was intended to capture **role-specialized functional separation**, rather than strict architectural independence. System 1 and System 2 use **separate LoRA adapters**, follow different **input-output specifications**, are trained with different **objectives**, and operate at different **timescales**.
> > >
> > > In particular, some recent work uses multi-agent for high-/low-level role decompositions that are primarily functional rather than fully architecturally independent [1,2]. We do not claim that this is the only valid framing, but rather that such usage exists in the literature.
> > >
> > > [1] Z. Wan, et al., "ReMA: Learning to Meta-Think for LLMs with Multi-Agent Reinforcement Learning," in *NeurIPS*, 2025.
> > >
> > > [2] H. Hsu, et al., "DEPART: Hierarchical Multi-Agent System for Multi-Turn Interaction," in *NeurIPS Multimodal Algorithmic Reasoning Workshop*, 2025.
> > >
> > > ---
> > >
> > > **Final remark**
> > >
> > > We hope this clarifies our position. If these clarifications address the reviewer’s concerns, we would appreciate reconsideration of the score. We would be happy to clarify any remaining points if helpful.

---

### Official Review · Reviewer_Tcya · 2026-03-13

**Soundness:** 3
**Presentation:** 3
**Significance:** 3
**Originality:** 3
**Overall Recommendation:** 5
**Confidence:** 3

**Summary:**

This paper proposes Multi2, a hierarchical framework for long-horizon interactive LLM agents. The method decomposes the agent into two roles: System 1, a high-level planner trained with supervised fine-tuning to generate context-aware sub-goals, and System 2, a low-level executor trained with an offline-to-online reinforcement learning pipeline to perform atomic actions. The framework aims to mitigate objective drift and improve token efficiency during multi-turn interaction. Experiments on ScienceWorld, ALFWorld, and TextCraft with multiple backbone models show consistent improvements over prompt-based and fine-tuning-based baselines. Overall, the authors explore an important problem in building reliable long-horizon LLM agents.

**Compliance With Llm Reviewing Policy:**

Affirmed.

**Final Justification:**

The authors’ rebuttal was clear and well-structured, and it effectively addressed my main concerns. As a result, I have decided to increase my score.

**Key Questions For Authors:**

Key Questions For Authors
1.Component contribution.
Multi2 introduces several changes simultaneously (e.g., separate LoRA adapters, SFT+RL training, new RL loss). Could the authors provide additional ablations to clarify which components contribute most to the performance gains? This would strengthen the causal claims of the paper.
2.Sub-goal completion mechanism.
How is sub-goal completion detected during execution, and how sensitive is the framework to errors in this signal? Clarifying this would help assess the robustness of the hierarchical design.
3.Online RL cost and stability.
Could the authors report more details on the online RL stage, such as interaction steps, training time, and stability across random seeds? This would help evaluate the practicality of the approach.

**Limitations:**

yes.

**Strengths And Weaknesses:**

Strengths：The paper studies an important problem in LLM agents: reliable long-horizon interaction, where issues such as objective drift and token inefficiency commonly arise. The proposed framework is clear and intuitive, separating planning (System 1) and execution (System 2) with role-aligned training (SFT for planning and RL for execution). The experimental evaluation is reasonably comprehensive, including multiple environments, backbone models, ID/OOD splits, token-efficiency analysis, and ablation studies. Additionally, the release of hierarchical datasets could be a useful resource for future research on long-horizon LLM agents.
Weaknesses：

1. Multiple design changes make attribution unclear.
Multi2 introduces several changes at once (e.g., separate LoRA adapters, SFT+RL training, new RL loss, and an offline-to-online pipeline), making it hard to isolate which components drive the gains. More fine-grained ablations would clarify this.

2. Limited analysis of online RL cost and stability.
The paper shows OOD improvements from online RL but provides little detail on interaction cost, training time, or stability across seeds.

3.The mechanism for sub-goal completion.
The mechanism for sub-goal completion is not clearly described; more details on how completion signals are defined and how the system handles incorrect detections would help clarify the robustness of the planner–executor interaction.

---

> ### Author Rebuttal · Authors · 2026-03-30
>
> We thank the reviewer for the important questions. The comments were helpful in clarifying several points and in strengthening the paper.
>
> ---
>
> ### **Could the authors provide additional ablations? (Q1, W1)**
> We agree that clearer attribution is important, since Multi² introduces several design choices simultaneously. To directly address this concern, we add finer-grained ablations and extend Section 5.7.
>
> - **Component-Wise Analysis**
>
> |Variant|Role-Specific Adapters|Role-Specialized Training|Hierarchical Structure|Proposed Offline Loss|ScienceWorld ID [%]|ScienceWorld OOD [%]|
> |:---|:---:|:---:|:---:|:---:|:---:|:---:|
> |Shared Adapter|✘|✔|✔|✔|45.57|22.93|
> |RL-SFT|✔|✘|✔|✔|52.05|18.15|
> |Only SFT|✔|✘|✔|—|53.72|18.35|
> |Only RL|✔|✘|✔|✔|57.32|28.93|
> |Single-Agent|—|—|✘|✔|56.02|26.39|
> |Vanilla-IQL|✔|✔|✔|✘|56.38|27.25|
> |**Proposed**|**✔**|**✔**|**✔**|**✔**|**60.68**|**29.04**|
>
> These results suggest that the gains are cumulative rather than attributable to a single factor. The largest drops come from removing **role-specific adapters** or the **role-specialized training strategy**, while the **hierarchical structure** and the **proposed offline loss** provide additional gains.
>
> - **Training Pipeline Analysis**
>
> |Variant|Offline RL|Online Adaptation|ScienceWorld ID [%]|OOD [%]|
> |:---|:---:|:---:|:---:|:---:|
> |Offline RL|✔|✘|60.68|29.04|
> |**Offline-to-Online RL**|**✔**|**✔**|**69.30**|**30.12**|
>
> Starting from the same offline model, adding the online adaptation further improves *ID* / *OOD* performance. Overall, the strongest performance comes from the **full combination** of parameter decoupling, role-specialized training, hierarchical structure, objective design, and online adaptation.
>
> ---
>
> ### **How is sub-goal completion detected during execution, and how sensitive is the framework to errors in this signal? (Q2, W3)**
>
> We agree that this mechanism should be clarified, since the completion signal is the key interface between the planner and executor.
>
> At training time, sub-goal completion is derived from the hierarchical trajectory construction: each low-level segment is associated with an active sub-goal, and completion supervision is derived from the parsed sub-goal boundaries in the dataset. At inference time, **System 2 outputs both an detailed action and a boolean completion signal** conditioned on the current sub-goal and observation. Control is returned to **System 1** only when System 2 predicts that the current sub-goal has been completed.
>
> To assess in practice, we analyze completion-signal errors on ALFWorld. We observe **no such cases on the *ID* split**, and only **1 case (1.47%) among 68 failed OOD episodes** in which System 2 emitted an incorrect completion signal. This suggests that completion-signal errors are **rare in our current setting** and do not appear to be a dominant source of failure. When such an error does occur, the framework can sometimes recover in subsequent planning steps because System 1 is re-invoked based on the updated interaction state.
>
> ---
>
> ### **Could the authors report more details on the online RL stage? (Q3, W2)**
> We agree that the practicality of the online RL stage should be assessed not only by its OOD gains, but also by its **interaction cost, training cost, and stability across random seeds**.
>
> - **Interaction Budget**
>
> In our online RL stage, System 2 interacts with the environment for up to **50 steps per episode**, with a total cost of *20,000 interaction steps*. In ALFWorld, successful episodes require only 10.15 ± 9.74 interaction steps on average (min=4, max=42), suggesting that the online phase is not dominated by excessively long trajectories.
>
> - **Training Time**
>
> Relative to offline training, the online RL stage adds only about **20% extra training time**. Despite this moderate overhead, it improves performance from the offline-only setting (60.68 / 29.04) to the online-adapted setting (69.30 / 30.12) on *ID* / *OOD*, indicating a favorable cost–performance trade-off.
>
> - **Stability across Seeds**
>
> We additionally summarize the online-stage results across **five random seeds** below.
>
> |ScienceWorld ID [%]|OOD [%]|
> |:---:|:---:|
> |69.30 ± 1.98|30.12 ± 0.96|
>
> These results show that the gains from online RL are reasonably stable across runs.
>
> - **Training-Curve Summary**
>
> To illustrate the stability over time, we report the training curve in ScienceWorld.
>
> |Timesteps|Performance [%]|
> |:---:|:---:|
> |0|61|
> |400|56|
> |800|59|
> |1200|64|
> |1500|68|
> |1800|69|
>
> This result suggests that online RL yields progressive refinement rather than only an early transient gain, while remaining practically manageable in our setting.
> We provide the seed-wise training curves on our project page https://anonymous-projectpage.github.io/Multi-Square/.
>
> ---
>
> **Final remark**
>
> We thank the reviewer again for the helpful and detailed questions. We hope that we have addressed all your comments, and we greatly appreciate your feedback.

---

> > ### Author Rebuttal · Reviewer_Tcya · 2026-04-02
> >
> > Thank you for your explanation.

---

> > > ### Author Response · Authors · 2026-04-03
> > >
> > > Thank you for your thoughtful follow-up. To address the remaining concern, we provide more detailed case analyses regarding sub-goal completion below.
> > >
> > > ---
> > >
> > > ## **Case Analyses and Explanations of Sub-Goal Completion**
> > > As described in Appendices A and C, at inference time, System 2 outputs both an **atomic action** and a **binary completion signal** (*True*/*False*) for the current sub-goal.
> > > More broadly, such control-signal mechanisms have been used in hierarchical LLM-agent frameworks [1,2].
> > > During training, the completion labels are derived deterministically from sub-goal boundaries in the System 2 dataset. This allows System 2 to learn not only which action should be taken to achieve a sub-goal, but also which action typically corresponds to the final step of completing that sub-goal.
> > >
> > > ### **Success Case Analysis**
> > > As illustrated in Appendix C.1, consider the ALFWorld task `Put a creditcard in shelf`.
> > >
> > > 1) Given the task, System 1 generates the sub-goal `Retrieve creditcard`.
> > > 2) System 2 receives the current observation along with the sub-goal, and outputs `Go to drawer 2; False`. This indicates that the current action does **not yet complete the retrieval sub-goal**.
> > > 3) Under the updated observation, System 2 then outputs `Open drawer 2; False`.
> > > 4) System 2 outputs `Take creditcard 2 from drawer 2; True`. This indicates that System 2 has learned to treat the successful acquisition of the target object as the **final action** that completes the retrieval sub-goal.
> > > 5) Once *True* is emitted, the current sub-goal is treated as completed and control returns to System 1, which generates the next sub-goal, `Move creditcard to shelf`.
> > >
> > > The table below summarizes this process, including the key inputs, outputs, and interactions.
> > >
> > > |Step|Decision Maker|Model Input|Model Output|Interaction|
> > > |:-:|:-|:-|:-|:-|
> > > |1|System 1|Task: `Put a creditcard in shelf`; Current observation|Sub-goal: `Retrieve creditcard`|System 1 → System 2|
> > > |2|System 2|Observation: `You are in the middle of a room. Looking quickly around you, you see a bed 1, ...`; Sub-goal|Action: `Go to drawer 2; False`|-|
> > > |3|System 2|Updated observation: `You arrive at drawer 2. The drawer 2 is closed.`|Action: `Open drawer 2; False`|-|
> > > |4|System 2|Updated observation: `You open drawer 2. The drawer 2 is open. In it, you see creditcard 2, ...`|Action: `Take creditcard 2 from drawer 2; True`|System 2 → System 1 (sub-goal completed)|
> > > |5|System 1|Updated observation: `You pick up the creditcard 2 from the drawer 2.`|Next sub-goal: `Move creditcard to shelf`|System 1 → System 2|
> > >
> > > Thus, the completion signal is generated by System 2 as part of its policy output. This signal determines whether execution should **continue under the current sub-goal** or **return control to System 1** for the next high-level decision.
> > >
> > > ### **Failure Case Analysis**
> > > To illustrate a completion-signal error, we analyze an *OOD* episode in which System 2 emits an incorrect completion signal.
> > >
> > > |Step|Decision Maker|Model Input|Model Output|Interaction|
> > > |:-:|:-|:-|:-|:-|
> > > |1|System 1|Task: `Examine an alarmclock with the desklamp`; Current observation|Sub-goal: `Retrieve alarmclock`|System 1 → System 2|
> > > |2|System 2|Observation: `You are in the middle of a room...`; Sub-goal|Action: `Go to desk 1; False`|-|
> > > |3|System 2|Updated observation: `You arrive at desk 1...`|Action: `Take alarmclock 1 from desk 1; True` | System 2 → System 1 (incorrectly emits *True*)|
> > > |4|System 1|Updated observation: `Nothing happens`; System 2's action trajectory|Sub-goal: `Retrieve alarmclock 2`|System 1 → System 2|
> > >
> > > In ALFWorld, object instances are indexed (e.g., `alarmclock 2`, `desk 1`), so selecting the correct index is critical for valid execution. In the task `Examine an alarmclock with the desklamp`, System 2 outputs `Take alarmclock 1 from desk 1; True`, although that object is not actually present there. The environment then returns `Nothing happens`, so the subsequent high-level decision is made from the updated state. This shows that completion-signal errors can occur, but their impact is limited by re-invocation under the new observation. In our *OOD* analysis, we observed this error in 1 out of 68 failed episodes (1.47%), suggesting that it is not a dominant failure mode in our current setting.
> > >
> > > [1] J. Choi, et al., "ReAcTree: Hierarchical LLM Agent Trees with Control Flow for Long-Horizon Task Planning," in *AAMAS*, 2026.
> > >
> > > [2] Z. Hu, et al., "Divide and Conquer: Grounding LLMs as Efficient Decision-Making Agents via Offline Hierarchical Reinforcement Learning," in *ICML*, 2025.
> > >
> > > ---
> > >
> > > **Final remark**
> > >
> > > We agree that failure-case analysis is an important part of understanding sub-goal completion. We thank the reviewer for this helpful suggestion, and we will revise the paper to include a clearer discussion of such cases.
> > >
> > > Once again, we sincerely appreciate the thoughtful questions. We hope that the clarifications above help address the remaining concern and will be considered in the final assessment.

---

### Official Review · Reviewer_MvPW · 2026-03-15

**Soundness:** 3
**Presentation:** 3
**Significance:** 2
**Originality:** 1
**Overall Recommendation:** 5
**Confidence:** 4

**Summary:**

This paper proposes $\text{Multi}^2$, a hierarchical training framework for LLM-based agents that separates decision-making into a high-level planner (System 1) and a low-level executor (System 2). The key idea is to apply role-specialized training objectives, where the planner is trained via supervised fine-tuning (SFT) to generate sub-goals, while the executor is optimized using an offline-to-online reinforcement learning pipeline to perform environment interactions. The proposed approach is evaluated on three interactive environments (ScienceWorld, ALFWorld, and TextCraft) across multiple backbone models, and the results show consistent improvements over several existing LLM-agent methods, including ReAct, Reflexion, ADaPT, GRPO, and the hierarchical baseline Glider.

**Compliance With Llm Reviewing Policy:**

Affirmed.

**Final Justification:**

My initial concerns included limited novelty, lack of stronger hierarchical baselines, missing multi-seed statistics, and unclear attribution of gains. The rebuttal effectively addressed most of these issues by adding stronger baselines, multi-seed results, and more detailed ablations, which significantly improve the empirical credibility of the work.

The main remaining concern is limited algorithmic novelty, as the contribution largely lies in combining existing components. Nevertheless, given the strengthened evaluation and clearer analysis, I find the work to be solid and valuable.

Overall, the rebuttal addressed most of my concerns and increased my confidence in the paper, so I have raised my score.

**Key Questions For Authors:**

- How much of the performance gain comes from the hierarchical architecture itself versus the RL training pipeline? The paper includes a useful ablation comparing different training assignments for System 1 and System 2 (RL-SFT, only RL, only SFT, and the proposed SFT-RL configuration), which helps validate the role-specialized training strategy. However, it is still unclear how much of the improvement comes from the hierarchical decomposition itself. For example, it would be helpful to compare against a single-agent baseline trained with the same offline-to-online RL objective.
- Why are stronger hierarchical RL baselines not included in the empirical comparison? The experiments mainly compare against Glider as the hierarchical baseline. Given that the paper emphasizes hierarchical training, it would be helpful to include comparisons with additional hierarchical RL or multi-turn RL training methods if feasible.
- Could the authors report multi-seed statistics for the main benchmark results? The main performance tables report point estimates only. Since long-horizon interactive environments can exhibit significant variance, reporting the mean and standard error across multiple seeds would improve the reliability of the empirical results.
- How sensitive is the framework to the quality of planner-generated sub-goals? Since System 2 depends on the sub-goals produced by System 1, it would be useful to better understand how errors in sub-goal generation affect overall performance.

**Limitations:**

No, the paper does not appear to include a dedicated discussion of limitations or potential negative societal impacts. While the proposed method focuses primarily on improving optimization robustness in RLHF settings, it would strengthen the paper to include a brief limitations section.

**Strengths And Weaknesses:**

**Strengths**
- The proposed decomposition between planning and execution is conceptually clear and aligns with the functional roles of these components. Training the planner with SFT and the executor with RL is a reasonable design choice given that planning primarily involves semantic reasoning while execution requires interaction with the environment.
- The method is evaluated across three interactive environments and multiple backbone models. In many settings the proposed approach improves over baselines, including Glider, which is one of the most closely related hierarchical LLM agent methods.
- The paper includes several ablation experiments examining training assignment, loss design, and online RL adaptation. In particular, the training assignment ablation (SFT planner + RL executor vs alternative assignments) directly tests the core design hypothesis of the paper.

**Weaknesses**
- (Limited algorithmic novelty) The main components of the method—hierarchical decomposition, SFT-based planning, offline RL training, and KL-regularized online RL—are individually well established. The novelty of the work primarily lies in the integration of these components rather than in the introduction of fundamentally new learning algorithms.
- (Limited comparison with other hierarchical RL approaches) Although the paper compares against several LLM-agent baselines, the comparison with other hierarchical RL approaches is limited. The experiments mainly compare against Glider as the hierarchical baseline, but including additional one or two more hierarchical RL or multi-turn RL training approaches would provide a stronger context for the proposed method.
- (Limited statistical robustness reporting) The main performance tables report point estimates only. Given the stochastic nature of long-horizon interactive environments, reporting multi-seed results with mean and standard error would strengthen the reliability of the empirical claims.
- (The effect of hierarchy itself is not fully isolated) The proposed method combines several components simultaneously: hierarchical decomposition, role-specialized training, separate adapters, and an offline-to-online RL pipeline. While the ablation studies are helpful, they do not fully isolate how much improvement comes specifically from the hierarchical design itself.

---

> ### Author Rebuttal · Authors · 2026-03-30
>
> We thank the reviewer for their thoughtful comments. Below, we describe how we have revised the paper to address the reviewer's questions.
>
> ---
>
> ### **Framework-Level Contribution (W1)**
>
> Our contribution is at the **framework and objective-design level**: hierarchical structure with **role-specific training**, where System 1 is trained with SFT and System 2 with an offline-to-online RL pipeline.
>
> Offline, the actor combines advantage-weighted behavior cloning with a *policy-anchored advantage term* (Eq. 6), preserving dataset-supported imitation while **encouraging actions with higher critic-estimated advantage**. Online, we further stabilize adaptation by *KL-regularizing the online policy toward the offline-trained policy* (Eq. 7), improving interaction-driven adaptation while discouraging abrupt policy shifts.
>
> As supported in Section 5.7, these objective-level choices are **consequential rather than incidental**: the offline objective improves over vanilla IQL (60.68% vs. 56.38%), and the online objective improves over vanilla AWAC (69.30% vs. 58.94%). Additional ablation evidence supporting this point is provided in our response to Reviewer Tcya (W1).
>
> Our claim is therefore *not* novelty at the level of individual algorithmic primitives, but rather that the combination of hierarchical role separation, role-aligned training, and objective-level design forms a principled framework for mitigating objective drift and improving execution robustness.
>
> ---
>
> ### **How much of the performance gain comes from hierarchical architecture?  (Q1, W4)**
>
> We thank the reviewer for this helpful suggestion. To address this, we compare against a flat *single-agent baseline* with the same offline-to-online RL objective.
>
> |Framework|ScienceWorld ID [%]|OOD [%]|
> |:-|:-:|:-:|
> |Single-Agent|59.64|28.92|
> |**Hierarchical (Proposed)**|**69.30**|**30.12**|
>
> The hierarchical framework consistently outperforms the single-agent baseline across all benchmarks, suggesting that **explicit hierarchy provides benefits beyond the RL pipeline**.
>
> ---
>
> ### **Why are stronger hierarchical RL baselines not included? (Q2, W2)**
>
> We agree that a broader empirical context beyond Glider is important. We additionally compare against AgentGym-RL [1], a *multi-turn RL* baseline, and ArCHer [2], a *hierarchical multi-turn RL* baseline.
>
> - **Comparison with Multi-Turn RL**
> |Method|Task Category [%]|Measure|Test-Cond|Find|Chem-Mix|Lifespan|
> |:-|:-|:-:|:-:|:-:|:-:|:-:|
> |AgentGym-RL [1]||20.75|28.92|0.00|0.00|66.67|
> |**Proposed**||**36.05**|**42.68**|**79.44**|**27.56**|57.73|
>
> - **Comparison with Hierarchical RL**
> |Method|ScienceWorld ID [%]|OOD [%]|
> |:-|:-:|:-:|
> |ArCHer [2]|39.89|18.96|
> |**Proposed**|**60.68**|**29.04**|
>
> These results show that our gains are *not specific to comparisons with Glider*, but persist against broader **multi-turn and hierarchical RL baselines**.
>
> [1] Z. Xi, et al., “AgentGym-RL: Training LLM Agents for Long-Horizon Decision Making through Multi-Turn Reinforcement Learning,” in *ICLR*, 2026.
>
> [2] Y. Zhou, et al., “ArCHer: Training Language Model Agents via Hierarchical Multi-Turn RL,” *arXiv preprint arXiv:2402.19446*, 2024.
>
> ---
>
> ### **Could the authors report multi-seed statistics? (Q3, W3)**
>
> Thank you for this helpful suggestion. We report the main benchmark results over **five random seeds** as mean ± one standard deviation.
>
> |Base Model|Method|ScienceWorld ID [%]|OOD [%]|ALFWorld ID [%]|OOD [%]|TextCraft Success [%]|
> |:-|:-|:-:|:-:|:-:|:-:|:-:|
> |Qwen-2.5 3B|Glider|54.69 ± 1.21|17.75 ± 4.07|52.86 ± 1.43|41.43 ± 1.07|18.50 ± 4.00|
> ||Proposed|**60.68 ± 0.41**|**29.04 ± 3.43**|**61.43 ± 6.45**|**49.29 ± 2.40**|**28.50 ± 5.20**|
> |Mistral 7B|Glider | 58.33 ± 0.22|28.22 ± 3.63|45.00 ± 2.69|45.71 ± 2.00|28.50 ± 5.75|
> ||Proposed|**69.97 ± 2.47**|**31.32 ± 0.68**|**56.43 ± 2.69**|**50.71 ± 3.60**|**44.50 ± 2.00**|
> |Llama-3.1 8B|Glider|60.48 ± 0.73|34.36 ± 2.07|43.57 ± 0.36|37.86 ± 3.57|9.50 ± 6.50|
> ||Proposed|**67.61 ± 1.27**|30.68 ± 4.47|**57.86 ± 1.48**|**56.43 ± 3.74**|**35.60 ± 3.34**|
>
> Multi² remains stronger than Glider in a large majority of benchmark/model settings. We will include full multi-seed results in the appendix.
>
> ---
>
> ### **How sensitive is the framework to planner-generated sub-goals? (Q4)**
>
> We agree that this is an important question, since System 2 is conditioned on sub-goals generated by System 1. We analyze this on ALFWorld by separating **planner-related failures** from other failed episodes.
>
> |Split|Planner-Related Failures|Total Failed Episodes|
> |:-|:-:|:-:|
> |ID|2 (3.70%)|54|
> |OOD|4 (5.88%)|68|
>
> Planner-related failures are relatively rare (2/54 on *ID* and 4/68 on *OOD*), suggesting that sub-goal errors are **not the dominant source of failure** in this setting.
>
> ---
>
> **Final remark**
>
> We sincerely appreciate the reviewer’s thoughtful comments. We hope that our clarifications and additional experiments have adequately addressed the concerns raised.

---

> > ### Author Rebuttal · Reviewer_MvPW · 2026-04-04
> >
> > I appreciate the authors' efforts to address my questions and concerns. The additional experiments and discussions have mostly resolved all of my concerns. The concern regarding limited algorithmic novelty still remains to some extent, as the contribution is primarily at the level of integrating existing components rather than introducing fundamentally new methods. However, given the strengthened empirical evidence and clearer attribution of gains, I believe the work is technically solid and provides meaningful insights. Based on the improvements made in the rebuttal, I am raising my score.

---

> > > ### Author Response · Authors · 2026-04-05
> > >
> > > We sincerely appreciate your thoughtful feedback and continued engagement with our paper. Your suggestions were highly valuable in improving the paper’s clarity and empirical support, and our discussion with you was very helpful in clarifying several important points and further strengthening the paper. We are also grateful for your consideration in raising the score.

---

### Decision · Program_Chairs · 2026-04-30

**Decision:**

Accept (regular)

**Comment:**

The paper proposes a hierarchical decomposition for long-horizon LLM agent tasks, with a planner trained via SFT and an executor trained via offline-to-online RL. The rebuttal was thorough and effective — it added adapter ablations, multi-seed statistics across all benchmarks, cross-dataset experiments, Q-value diagnostics, and comparisons against two additional baselines (AgentGym-RL, ArCHer), resolving the main technical concerns raised by the panel. Two reviewers raised their scores in response, and a third acknowledged that technical issues were fully addressed. The long-horizon stability results are the paper's strongest empirical finding: performance remains robust as horizon increases, unlike baselines which degrade substantially. The released hierarchical datasets across three benchmarks are also a valuable community contribution. The main limitation, noted by all four reviewers including the supporters, is that the individual components (hierarchical decomposition, SFT planning, IQL-based offline RL, KL-regularized online adaptation) are well-established — the contribution is in their principled integration rather than in new algorithmic primitives. This is a real concern, but given the strength of the empirical evidence, the quality of the rebuttal, and the practical value of the long-horizon robustness results, I find the overall contribution sufficient for acceptance as a systems-level advance.